# Genome-Wide Identification of DREB Gene Family in Kiwifruit and Functional Characterization of Exogenous 5-ALA-Mediated Cold Tolerance via ROS Scavenging and Hormonal Signaling

**DOI:** 10.3390/plants14162560

**Published:** 2025-08-17

**Authors:** Ping Tian, Daming Chen, Jiaqiong Wan, Chaoying Chen, Ke Zhao, Yinqiang Zi, Pu Liu, Chengquan Yang, Hanyao Zhang, Xiaozhen Liu

**Affiliations:** 1Key Laboratory for Forest Resources Conservation and Utilization in the Southwest Mountains of China, Ministry of Education, Southwest Forestry University, Kunming 650224, China; yiwangbaoyq@swfu.edu.cn (P.T.); wjqlzp522624@swfu.edu.cn (J.W.); yxy1999@swfu.edu.cn (C.C.); zhaoke@swfu.edu.cn (K.Z.); ziyinqiang@swfu.edu.cn (Y.Z.); 19143082025@163.com (C.Y.); 2Institute of Tropical Ecological Agriculture, Yunnan Province Academy of Agricultural Sciences, Chuxiong Yi Autonomous Prefecture 675000, China; cdming69@163.com; 3Anhui Key Laboratory for Horticultural Crop Quality Biology, College of Horticulture, Anhui Agricultural University, Hefei 230036, China; puliu@ahau.edu.cn

**Keywords:** kiwifruit, DREB gene family, exogenous 5-Aminolevulinic acid, low-temperature stress

## Abstract

Dehydration response element binding proteins (DREBs) have been identified as major regulators of cold acclimatization in many angiosperms. Cold stress is one of the primary abiotic stresses affecting kiwifruit growth and development. However, kiwifruit is currently one of the most widely consumed fruits worldwide because of its high nutritional value. 5-Aminolevulinic acid (5-ALA) is a nonprotein amino acid known for its distinct promotional effects on plant resistance, growth, and development. However, studies on the function of the kiwifruit DREB gene in alleviating low-temperature stress in its seedlings via exogenous 5-ALA have not been reported. Therefore, in this study, we performed a genome-wide identification of DREB gene family members in kiwifruit and analyzed the regulatory effects of exogenous 5-ALA on kiwifruit DREB genes under low-temperature stress. A total of 193 DREB genes were identified on 29 chromosomes. Phylogenetic analysis classified these genes into six subfamilies. Although there were some differences in cis-elements among subfamilies, all of them contained more biotic or abiotic stresses and hormone-related cis-acting elements. GO and KEGG enrichment analyses revealed that AcDREB plays an essential role in hormone signaling, metabolic processes, and the response to adverse stress. Under low-temperature stress, the application of exogenous 5-ALA inhibited the accumulation of APX and DHAR, promoted an increase in chlorophyll, and increased the accumulation of enzymes and substances such as 5-ALA, MDHAR, GR, ASA, GAH, and GSSH, thereby accelerating ROS scavenging and increasing the cold hardiness of kiwifruits. Functional analysis revealed that 46 differentially expressed DREB genes, especially those encoding *AcDREB69*, *AcDREB92*, and *AcDREB148*, which are involved in ethylene signaling and defense signaling, and, after the transcription of downstream target genes is activated, are involved in the regulation of low-temperature-stressed kiwifruits by exogenous 5-ALA, thus improving the cold tolerance of kiwifruits. Notably, *AcDREB69*, *AcDREB92*, and *AcDREB148* could serve as key genes for cold tolerance. This study is the first to investigate the function of AcDREB genes involved in the role of exogenous 5-ALA in regulating low-temperature stress, revealing the regulatory mechanism by which DREB is involved in the ability of exogenous 5-ALA to alleviate low-temperature stress.

## 1. Introduction

The growth and development of plants are affected by the interaction of internal and external factors, including water, light, low and high temperatures, and internal factors, including physiological and biochemical regulation, metabolic pathways, and signal transduction. Temperature is an essential factor affecting plant growth, development, and geographic distribution, and cold stress is one of the primary abiotic stresses affecting the development and growth of kiwifruit [1,2,3]. Prolonged low-temperature stress has adverse effects on plants, as manifested by a low germination rate, growth retardation, and even complete tissue or plant death [4]. In response to damage caused by low-temperature stress, plants have evolved an adaptive mechanism involving the upregulation of the antioxidant defense system. The antioxidant system comprises both antioxidant enzymes and antioxidant substances [5,6]. Low-temperature stress disrupts the original balance of the antioxidant system in plants, which can be detrimental to the growth and development of kiwifruit seedlings. Numerous studies have demonstrated that low temperature negatively impacts plant nutrient absorption and accumulation, chlorophyll content, photosynthetic capacity, oxidative stress, metabolic processes, the defense system, and hormonal balance [7]. For example, low-temperature stress leads to the accumulation of reactive oxygen species (ROS) and malondialdehyde (MDA) in plants, which disrupts the original dynamic equilibrium, leading to biofilms, the oxidation of proteins and nucleic acids, damage to plant tissues, or plant cell death [8,9]. Low temperature in cotton induces the expression of ICE2, and the overexpression of GthICE2 in *Arabidopsis* results in a lower MDA content and higher SOD and POD activities [10]. Studies have shown that after cold treatment (4 °C) for two days, the activities of the enzymes CAT, SOD, and POD increase, and more ROS accumulate in rice [11]. Therefore, in the study of the low-temperature response of plants, the antioxidant enzyme system is often used as an essential indicator to measure and identify the cold resistance of plants.

Dehydration response element binding proteins (DREBs) are members of the AP2/ERF transcription factor superfamily. This family contains a conserved AP2 domain that strongly interacts with dehydration response elements (DREs) (core motif A/GCCGAC) [12,13,14] and has been identified as a master regulator of cold adaptation in many angiosperms [15,16,17,18]. The DREB gene plays an essential role in plant hormone signaling pathways to protect against pathogens and abiotic stresses [19]. In *Arabidopsis*, the DREB subfamily can be divided into six subgroups, from A1 to A6 [15]. Members of the DREB A1 subfamily are sensitive to cold stress and regulate the expression of cold stress-related genes [20]. *Arabidopsis AtTINY* belongs to the DREB A3 family. The overexpression of *AtTINY* improves cold tolerance by increasing the expression of cold stress-related genes (such as COR6.6, COR15A, and COR78) [21]. *DREB1* and *DREB2* induce the expression of cold- and drought-responsive genes. This signal transduction pathway is vital for the response of plants to cold stress [22]. Other studies have shown that many other TFs (such as PIF, MPK, EIN, and CRPK) are also involved in DREB-dependent signaling processes, underscoring the importance of the DREB protein in plant cold adaptation [23]. Transgenic methods for DREB expression in crops such as tobacco, barley, tomato, and alfalfa can significantly increase plant growth and tolerance to cold stress [24]. One study revealed that, as an essential signaling integrator, CBF plays a key role in the early response of *Arabidopsis* to cold stress by activating target gene transcription [25]. AtCBF1, AtCBF2, and AtCBF3 are tandemly distributed on chromosome 4 of *A. thaliana*. Knockout of CBF1 or CBF3 can increase the sensitivity of downstream target genes to freezing stress [26]. Chen et al. isolated DREBs from soybeans via rapid amplification of cDNA ends and reported that *DREB3* is involved in the abscisic acid-independent cold stress-responsive signaling pathway [27]. The *GmDREB3*-overexpressing plants presented increased cold resistance. Under low-temperature stress, the fresh weight and osmotic pressure of these plants were greater than those of the wild-type plants. These results highlight the importance of the DREB gene family in the plant response to cold stress.

Moreover, plant growth regulators can relieve the damage caused by abiotic stress, increase stress resistance, and promote plant growth. Studies have shown that 5-aminolevulinic acid (ALA) is widely present in living cells of animals and plants, bacteria, and fungi and that it is a nonprotein amino acid that plays an essential role in chlorophyll, heme, and vitamins in organisms. Key precursors for the synthesis of cyclic tetrapyrrolyl compounds such as B12 [28] can not only improve the photosynthetic efficiency of plants but also increase cold resistance and low light tolerance and function similarly to plant growth regulators, significantly promoting plant stress resistance and growth and development [29]. Currently, 5-ALA has been used in apples [30], rice [31], Chinese yew [32], tea trees [33], thorn grapes [34], and Feng dan [35] and has effects on low temperature, germination, anthocyanin synthesis, and photosynthesis; however, few studies have investigated the application of 5-ALA in kiwifruits, especially in kiwifruit seedlings, in response to low-temperature stress. The effect and method of its application, appropriate concentration, and mechanism are still unclear.

Kiwifruit contains a large amount of vitamin C, is edible, and is one of the most consumed fruits in the world. Fresh fruit and processed products have become commodities in domestic and international markets and have high economic value. Anthocyanins [36], flavonoids [37], and many fat-soluble terpenoids may be the main components of TCMs and therefore have some medicinal value. Kiwifruit prefers to grow in relatively humid and sheltered environments, where the average temperature in the winter is not lower than 0 °C, and the temperature in the summer should not be higher than 40 °C during the annual frost-free period; therefore, in the winter, spring, or autumn, continuous low temperatures damage the growth of kiwifruits, making it impossible for them to overwinter safely. Studies have shown that 5-ALA is a ubiquitous nonprotein amino acid and a key precursor for the biosynthesis of all tetrapyrrole substances, and it has a significant effect on improving photosynthetic efficiency and cold resistance in plants. Investigating the physiological characteristics and molecular response mechanisms of kiwifruit seedlings under the alleviation of low-temperature stress by exogenous 5-ALA is essential for addressing problems related to the growth and actual production of kiwifruit. In this study, the identification, analysis of physicochemical properties, phylogenetic analysis, chromosomal mapping, identification of cis-regulatory elements, and GO and KEGG enrichment analyses of the DREB gene family were used to preliminarily elucidate the functions of the DREB family genes in kiwifruits. In addition, this study aimed to further investigate the expression pattern of the DREB gene family involved in the participation of exogenous 5-ALA in the relief of kiwifruit from low-temperature stress. The purpose of this study was to analyze how exogenous 5-ALA regulates damage to kiwifruit seedlings under low-temperature stress through AcDREB to improve the cold resistance of kiwifruit, elucidate the mechanism of enhanced resistance, and verify the hypothesis that AcDREBs are key response factors through which exogenous 5-ALA increases cold resistance in kiwifruits. This study can also provide protective measures to reduce harm to kiwifruit under low-temperature stress and has an essential guiding role in the breeding of cold-tolerant kiwifruit in the future. This study also provides a theoretical and practical foundation for improving plant cold tolerance and enhancing plant growth adaptability.

## 2. Results

### 2.1. Physicochemical Properties of Kiwifruit DREB Family Genes

The final 193 DREB genes were annotated according to chromosomal location and named *AcDREB1*–*AcDREB193*. Basic parameters were analyzed to explore the functions of the DREB family (Figure 1, Appendix A). The number of amino acids in the DREB family proteins ranged from 81 (*AcDREB131*) to 922 (*AcDREB50*), the isoelectric points were between 4.51 (*AcDREB80*) and 10.71 (*AcDREB66*), and the molecular weights were in the range of 9.24 kDa (*AcDREB131*) to 102.63 kDa (*AcDREB50*). The instability coefficient varied between 28.20 (*AcDREB131*) and 219.44 (*AcDREB162*), indicating that the members originated from different physiological environments. More than 55.259% of the kiwifruit DREB family proteins were weakly acidic because their pI was <7. The GRAVY values were negative, ranging from −1.148 (*AcDREB10*) to −0.112 (*AcDREB46*), suggesting hydrophilicity. The aliphatic indices ranged from 37.99 (*AcDREB157*) to 89.37 (*AcDREB135*). Subcellular localization analysis revealed that 125 AcDREB family proteins were located in the nucleus; 53 AcDREB family proteins were present in both the cytoplasm and nucleus; 11 AcDREB family proteins were located in the cytoplasm; the *AcDREB7* and *AcDREB130* genes were located in the chloroplast; and the *AcDREB54* gene protein was located in the cytoplasm, mitochondria, and nucleus. The *AcDREB104* gene protein is located in the cytoplasm and mitochondria.

### 2.2. Phylogenetic Analysis of the Kiwifruit DREB Family Genes

To explore the phylogenetic relationships of the AcDREB gene, 193 AcDREBs and 56 AtDREBs were compared and processed via the maximum likelihood (ML) method with appropriate default parameters set to construct a phylogenetic tree (protein sequences are shown in Appendix A). The 193 DREB genes in kiwifruit can be divided into six subgroups (A1–A6) (Figure 2). A4 and A3 were the largest and smallest subgroups, with 128 and 2 kiwifruit DREB genes, respectively. These results indicated that the DREB genes of kiwifruit and *Arabidopsis* were unevenly distributed among all the subgroups. Because specific AP2s recognize and bind to the cis-regulatory sequences of DREs (dehydration response elements) in the regulatory network, this type of AP2 domain represents the typical characteristics of DREB genes. Therefore, in this study, we used a dataset of 193 AcDREB genes to perform a preliminary analysis of the response of Actinidia chinensis to cold stress.

### 2.3. Structure and Conserved Domain Analysis of the Kiwifruit DREB Gene

To explore the distribution and structural diversification of the conserved motifs in AcDREB proteins, we used the MEME online tool to process a maximum of 10 motifs via in silico prediction. The motifs were renamed according to numbers from 1 to 10 (Figure 3). The results revealed that conserved motifs 1, 2, and 5 were highly conserved (Figure 3A). However, the positions of the same motifs in different protein sequences vary, which might be related to the structure and function of the proteins. Motif 10 appeared only in DREB genes 25, 72, 74, 80, 86, 104, 142, 154, and 171, indicating that these genes may be involved in specific functions. In contrast, motif 1 was characterized in all the AcDREBs, indicating that this motif is highly conserved.

The distribution of conserved motifs among the AcDREB gene family may also be affected by gene structure. The distribution of gene structure is key to studying evolution within a gene family. The results of the gene structure analysis of the 193 AcDREB genes (Figure 3D) revealed that of the 193 kiwifruit DREB genes, the majority (192) were within 19 kb in length. The length of the remaining gene (*AcDREB50*) is approximately 22 kb. The number of introns and exons of the kiwifruit DREB gene was quite different; the number of introns ranged from 0 to 16, and the number of exons ranged from 1 to 17. Although the *AcDREB50* gene was the longest, it was not the gene with the highest number of introns and exons. The gene with the most introns and exons was *AcDREB59*. In summary, many differences in the number and content of motifs between the different subfamilies of AcDREBs were revealed, which may be related to differences in their gene structures and less variation.

### 2.4. Analysis of Cis-Acting Elements in Gene Promoters

In the promoter sequences of transcription factors, various cis-regulatory elements act as key regulators of gene expression. The Plant CARE program was used for the 2000 bp upstream region of 193 AcDREB genes to analyze potential cis-regulatory elements involved in transcriptional activity and response to biotic and abiotic stresses in kiwifruits. The DREB gene plays a crucial role in controlling the fundamental processes of plant growth and development. Furthermore, it exhibits extraordinary versatility in its ability to respond to many different plant hormones and environmental stresses. Cis-acting element analysis was performed on the promoters of the DREB gene family in kiwifruits (Figure 4, Appendix A). The results revealed that the promoter regions of the A1–A6 subfamily genes all contained cis-acting elements related to abiotic and biotic stresses, the light response, hormones, and growth and development. Among the cis-elements involved in the response to abiotic and biotic stresses, except for the cis-acting elements (AREs) involved in hypoxia-specific induction, they were not present in the A1, A3, and A6 subfamilies, and the A3 subfamily did not have low temperature- or drought-related factors. In addition to the cis-acting elements of the inducible MYB-binding locus, each subfamily contained ARE, CCAAT-box, GC-motif, LTR, MBS, and TC-rich repeat cis-acting elements (Figure 4A). These findings indicate that the DREB gene family plays a crucial regulatory role in adverse stress conditions. Among the cis-elements involved in the light response, the A3 subfamily did not have the GATA motif or MRE cis-elements. The remaining subfamilies were associated with the GATA motif, GT1 motif, MRE, and Sp1 cis-acting elements (Figure 4B). In addition, among the cis-elements associated with phytohormones, all subgroups contained ABRE, CGTCA-motif, GARE-motif, MBSI, P-box, TATC-box, TCA-element, TGACG-motif, and TGA-element cis-acting elements except for subgroup A3, which had no TATA-box or MBSI-elements, and subgroup A5, which had no GARE-motif-elements (Figure 4C). In particular, ABREs, the CGTCA-motif, and the TGACG-motif were relatively more abundant, which was related to the regulation of low-temperature stress by DREB. Among the cis-elements involved in plant growth and development (Figure 4D), the CAT-box, CCAAT-box, circadian, GCN4_motif, and O2-site subgroups contained many DREB genes, indicating that DREB genes in kiwifruits are associated with O2-site, CAT-box, GCN4_motif, circadian, and CCAAT-box components. Subgroup A6 was not related to the CAT-box components. Interestingly, only O2 sites and CAT boxes were present in subgroup A3, which may be associated with the number of genes and functional specificity of the genes in this subgroup. Notably, among the differentially expressed genes of the kiwifruit DREB gene (Figure 4E), *AcDREB82* contained the most cis-acting elements, *AcDREB169* contained the fewest, and only MeJA-responsive motif and GC-motifs were involved. Thus, *AcDREB169* may play a crucial regulatory role in the MeJA response. Most DEGs were related to hormones, such as abscisic acid, MeJA, salicylic acid, and gibberellin. Moreover, many DEGs were also found to be involved in cis-acting elements involved in abiotic versus biotic stress responses, and they were related to AREs, TC-rich repeats, and LTR- and GC-motif elements.

In conclusion, the number and location of cis-acting elements are closely related to gene function. In addition, there were more abiotic and biotic stress-related and hormone-related cis-elements, especially cis-elements such as AREs, CCAAT boxes, LTRs, MBSs, ABREs, CGTCA-motifs, TGACG-motifs, TCA-elements, and TGA-elements. This functional element (Figure 4) is associated with the regulatory role of this gene family in stress. The *AcDREB69* gene is involved in response elements such as the GT1-motif, MRE, Sp1, ABRE, CGTCA-motif, ERE, MBSI, TCA-element, TGACG-motif, ARE, CCAAT-box, GC-motif, LTR, and GCN4-motif. For *AcDREB92*, the genes were involved in response elements such as MRE, ABRE, ERE, ARE, and CAT-box; the *AcDREB148* gene was involved in response elements such as the GT1-motif, Sp1, CGTCA-motif, ERE, TGACG-motif, ARE, GC-motif, LTR, and MBS (Figure 4E). These three genes are DEGs in the KEGG pathway, and they all participate in the ERE and ARE response elements. These two response elements are classified as abiotic and biotic stress, indicating that the EREs and ARE response elements jointly participated by these three genes may be the primary regulatory elements of this gene family in cold tolerance, and also suggesting that these three genes may be the primary regulatory genes in cold tolerance.

### 2.5. Chromosomal Mapping and Collinearity Analysis of the Kiwifruit DREB Gene Family

To study the genomic distribution of DREB genes on kiwifruit chromosomes, the physical locations of all DREB genes in kiwifruit were determined via the location data of DREB genes obtained from the kiwifruit genomics database. Moreover, the gene duplication events of whole-genome duplication (WGD), segmental duplication, and/or tandem duplication were analyzed to study the expansion mode of the DREB gene family in kiwifruit species. Based on the chromosomal gene location information, the distribution of the 193 identified AcDREB genes on the 29 chromosomes was uneven (Figure 5). Chr4, Chr7, and Chr24 had the highest number of AcDREB genes (12), whereas Chr29 had the least AcDREB genes (2). Interestingly, although Chr 1 is the longest chromosome, it does not have many DREB members (eight), indicating a lack of significant correlation between chromosome length and the distribution of DREB genes.

Gene collinearity analysis of the DREB gene family revealed 183 pairs of segmental duplicates (Figure 5), with the highest number of segmental duplicates on chromosome 4 (26 pairs), followed by chromosome 7 (22 pairs) and duplicate pairs of genes on all remaining chromosomes. These results indicate that fragmental repeats play a vital role in the AcDREB family expansion. Among the 183 pairs of segment repetitive genes, *AcDREB69*, *AcDREB92*, and *AcDREB148* were DEGs that appeared in the KEGG pathway; *AcDREB69* and *AcDREB92* each had one segment duplication, and *AcDREB148* had two segment duplications (Appendix A).

To increase the understanding of genetic differentiation, gene duplication, and evolutionary patterns in the DREB gene families of *Arabidopsis*, apple, rice, and kiwifruit, their homologous DREB genes were analyzed (Figure 6). The results revealed 344 homologous gene pairs between kiwifruit and apples. Among them, there were relatively more homologous genes on chromosomes 1, 4, 7, 16, 23, and 24, with a higher occurrence frequency. There were 262 homologous gene pairs between kiwifruit and *Arabidopsis* and 111 homologous gene pairs between kiwifruit and rice. These results suggest that, compared with those of monocots, the evolutionary relationships between kiwifruit and the dicot apple are closer, which is consistent with the results of previous studies on BAM genes between kiwifruit and apples [38].

### 2.6. GO and KEGG Enrichment Analyses of the Kiwifruit DREB Gene Family

To further elucidate the biological functions of the 193 AcDREB genes, the proteins of their genes were annotated for gene function via the EggNOG-MAPPER database. Among the 193 genes, 149 kiwifruit DREB genes were annotated and assigned to three categories: biological process (BP), molecular function (MF), and cellular component (CC) (Figure 7). In terms of molecular function, the enrichment of AcDREB genes, such as transcription regulatory activity, DNA binding transcription factor activity, small-molecule binding, and organic cyclic compound binding, was high, indicating that the molecular function of the AcDREB gene family plays a crucial role in regulation. Among the enriched cellular components, the number of AcDREB genes enriched in senescent intracellular parts, intracellular anatomical structures, and organelles was greater. In particular, most genes were enriched in biological processes and played an essential role in the AcDREB gene family regulation. The synthesis and metabolic processes of organic substances and environmental stimuli-responsive and hormone-stimuli-responsive or hormone-mediated signaling pathways were enriched with more AcDREB genes. The synthesis and metabolic processes of organic matter were enriched, e.g., the regulation of macromolecule metabolic processes, metabolic processes, and primary metabolic processes. In response to environmental stimuli such as cold stress, water deprivation, and temperature stimulation, hormone-related signals include ethylene-activated signaling pathways, abscisic acid-activated signaling pathways, and the jasmonic acid response, indicating that the AcDREB gene may adapt plants to different environments by regulating various plant parts.

In addition, KEGG enrichment analysis (Appendix A) revealed that only 36 of the 193 kiwifruit DREB genes were enriched in the plant hormone signal transduction (ko04075), MAPK signaling pathway–plant (ko04016), plant–pathogen interaction (ko04626), and spliceosome (ko03040) pathways. Among them, eight genes were involved in both signal transduction pathways and the MAPK signaling pathway–plant, nineteen genes were enriched in plant–pathogen interaction pathways, and nine genes were enriched in zygosomes. Interestingly, the genes enriched in zygosomes, signal transduction pathways, and the MAPK signaling pathway in plants all belong to the A-4 subgroup, which may be related to the specific function of this subfamily. The genes enriched in the plant–pathogen interaction pathway were distributed in the A-6 subgroup and the A-4 and A-5 subgroups, which might be related to the enriched functions of the genes.

### 2.7. Expression Pattern of the AcDREB Gene Family in Kiwifruit Under Exogenous 5-ALA-Regulated Low-Temperature Stress

#### 2.7.1. Effects of Exogenous 5-ALA on the Physiological Indicators of Kiwifruit Seedlings Under Low-Temperature Stress

To understand the potential biological function of kiwifruit DREB in low-temperature stress regulation by exogenous 5-ALA, kiwifruit seedlings were grown for seven days under normal temperature and constant light (CK), low-temperature stress (LT) at −2 °C, and root irrigation with 3 g/L 5-ALA solution (LT-5-ALA) before low-temperature stress. Compared with those of the CK seedlings, the petioles of the kiwifruit seedlings were soft after low-temperature stress at −2 °C for five hours, the degree of wilting of the leaves continued to increase, a curling phenomenon appeared, and the apical leaves experienced significant frostbite. These symptoms were significantly relieved after the patients received 3 g/L 5-ALA (Figure 8A). Compared with that in the CK treatment, the 5-ALA content in the leaves significantly differed, while the 5-ALA content in the leaves after root irrigation with LT-5-ALA was 63.245% greater than that in the leaves in the LT treatment (Figure 8B). Under low-temperature stress, the Chl a and Chl b contents of the kiwifruit seedlings were greater than those under normal-temperature conditions. After the roots were irrigated with exogenous 5-ALA, the Chl a and Chl b contents were significantly greater than those under the CK and LT treatments, and the total chlorophyll content of the kiwifruit seedlings irrigated with exogenous 5-ALA was significantly greater than that under the CK and LT treatments (Figure 8C). Compared with those under CK, the enzyme activities of DHAR and MDHAR under low-temperature stress were significantly lower. However, their contents slightly increased after exogenous 5-ALA application (Figure 8D,E). The expression level of GR activity in the CK treatment group was low. However, after low-temperature treatment at −2 °C for five hours, the expression level of GR activity in the LT treatment group was significantly greater than that in the CK treatment group, with an increase of 45.28%. Low-temperature stress led to increased GR activity in kiwifruit seedling leaves. The application of exogenous 5-ALA under low-temperature stress further increased the GR activity; the GR activity of the LT-5-ALA treatment group was significantly greater than that of the CK and LT treatment groups, with increases of 72.27% and 18.58%, respectively (Figure 8F). Compared with those of the LT and LT-5-ALA groups, the APX activity levels of the seedling leaves of the CK treatment group were significantly greater than those of the LT and LT-5-ALA groups. However, APX activity did not increase after low-temperature stress and was lower than that in the LT treatment group (Figure 8G). Low-temperature stress significantly induced the accumulation of antioxidants (AsA, GSH, and GSSG) in the kiwifruit seedlings. Compared with those in the CK group, the AsA, GSH, and GSSG contents in the LT treatment group were significantly greater (Figure 8H–J). After the roots were irrigated with exogenous 5-ALA, the antioxidant contents further increased significantly (Figure 8H–J). These results suggest that under low-temperature stress, exogenous 5-ALA can significantly relieve the degree of wilting and frostbite of kiwifruit seedlings, induce the accumulation of endogenous 5-ALA in the leaves of kiwifruit seedlings, effectively relieve chlorophyll damage, and further enhance antioxidant effects. Enzyme (DHAR, MDHAR, and GR) activities and the contents of antioxidant substances (ASA, GSH, and GSSG) increased.

#### 2.7.2. Expression Profiling of the AcDREB Gene Family in Kiwifruit Under Exogenous 5-ALA-Regulated Low-Temperature Stress

To further understand the expression pattern of the AcDREB gene family when exogenous 5-ALA relieves low-temperature stress, this study analyzed the characteristics and transcriptome data of kiwifruit under control (CK), low-temperature stress (LT), and exogenous 5-ALA (LT-5-ALA) treatments before low-temperature stress. Among the 193 AcDREB gene family members identified, 170 genes were expressed (Figure 9A). The expressed AcDREB gene family transcription factors were divided into eight clusters according to their expression patterns (C1–C8). In particular, the greatest number of AcDREB genes (27) were expressed in C8, which tended to be upregulated from CK to LT and then to LT-5-ALA, and the greatest number of DEGs appeared in this pattern. This was followed by the C7 pattern, which revealed a more consistent trend in expression than did the C8 pattern. The significant expression trends among the three treatments in the C7 and C8 patterns indicated that the AcDREB genes in the C7 and C8 clusters play important roles in the regulation of exogenous 5-ALA in response to low-temperature stress. In contrast, the number of genes expressed in both C1 and C4 was 16, which was relatively low, and no DEGs existed in either pattern. The expression of C1 genes was greater in CK than in LT, whereas the expression patterns of LT and LT-5-ALA were similar. The expression of the C4 genes was greater in CK than in LT and lower in LT than in LT-5-ALA. Among the clusters of C1 to C8, the expression patterns of the C2 and C3 clusters are more consistent. The C6 pattern was the only clustering pattern that showed a sequential decrease in expression among the three treatments, indicating that some genes did not increase in expression in response to exogenous 5-ALA administration. The expression of the C5 pattern was downregulated from CK to LT and upregulated in response to LT-5-ALA, indicating that low-temperature stress downregulated the expression of the genes and that the application of exogenous 5-ALA could upregulate the expression of the genes in this pattern, thus alleviating the injury caused by low-temperature stress. However, among the 170 expressed AcDREB genes, 46 DEGs were identified, of which 1 DEG was in all three treatment groups (Figure 9B). In addition, compared with CK, LT-5-ALA had 38 DEGs, 34 of which were upregulated and 4 of which were downregulated (Figure 9C). Among these DEGs, *AcDREB167*, *AcDREB168, AcDREB173*, and *AcDREB182* were upregulated, and *AcDREB36*, *AcDREB53*, and *AcDREB187* were downregulated. The differential expression of *AcDREB51* was significant. Compared with those under CK, there were 26 DEGs under low-temperature stress (Figure 9C), 23 of which were upregulated and 3 of which were downregulated. The differential expression of the upregulated *AcDREB82*, *AcDREB167*, and *AcDREB168* genes and the downregulated *AcDREB71*, *AcDREB187*, and *AcDREB192* genes was significant. However, compared with LT, root irrigation with 3 g/L 5-ALA before low-temperature stress resulted in nine DEGs, including eight upregulated DEGs and one downregulated DEG (Figure 9C). The differential expression of upregulated *AcDREB146* and downregulated *AcDREB53* was significant. *AcDREB45* was a DEG common to the CK and LT-5-ALA treatments. Compared with the CK group, *AcDREB45* expression was upregulated in both the LT treatment group and after root irrigation with 5-ALA solution before low-temperature stress. These results suggest that root irrigation with 5-ALA solution did not significantly affect the expression of some genes before low-temperature stress. Among the 46 DEGs, 3 genes, *AcDREB69*, *AcDREB92*, and *AcDREB148*, appeared in the KEGG pathway. Among them, the expression of the genes *AcDREB69* and *AcDREB148* was significantly upregulated after LT-5-ALA treatment, and these two genes belonged to the C5 and C7 modes, respectively. The *AcDREB92* gene is associated with the C8 pattern, and its expression was significantly upregulated under both low-temperature stress and root-infused 5-ALA (Figure 9A).

#### 2.7.3. Functional Analysis of the AcDREB Gene Family in Kiwifruit Under Exogenous 5-ALA-Regulated Low-Temperature Stress

To further explore the functional information of the 46 DEGs related to low-temperature stress, we screened 10 functional entries related to stress or metabolism via GO enrichment analysis (Figure 10). The results revealed that in these functional entries, 39 DEGs of the AcDREB gene family were enriched, and most genes that exhibited versatility were involved in various functional responses. These results suggest that under cold stress, AcDREB may participate in the regulation of exogenous 5-ALA in the regulation of cold stress-induced kiwifruit injury through different pathways. Among them, the genes enriched in metabolic processes were the most enriched; the genes enriched in endogenous stimuli, signal transduction, the ethylene signal response, the defense response, and the oxygen and stress response were relatively more enriched; and the genes enriched in response to signaling pathways activated by abscisic acid were the least enriched. Compared with those in CK and under LT-5-ALA treatment, 30 DEGs were enriched, with 26 upregulated genes, 4 downregulated genes, and 4 genes enriched in the KEGG pathway (Figure 10A). Compared with those in the CK treatment, 22 DEGs were enriched under the LT treatment, with 19 upregulated, 3 downregulated, and 2 enriched in the KEGG pathway (Figure 10B). However, compared with those under LT, fourteen DEGs under LT-5-ALA treatment were enriched, with nine upregulated, five downregulated, and two enriched in the KEGG pathway (Figure 10C). Compared with those in the CK treatment group, more DEGs in the LT-5-ALA treatment group than in the LT group were differentially expressed. However, although relatively fewer DEGs were differentially expressed after LT-5-ALA treatment than after LT, new DEGs appeared after LT-5-ALA treatment compared with those after LT treatment. However, new DEGs appeared after LT-5-ALA treatment but not after LT treatment. These results indicated that under LT-5-ALA treatment, the novel differentially expressed AcDREB gene was involved in the regulation of low-temperature stress-induced damage to kiwifruit seedlings. *AcDREB46*, *AcDREB69*, and *AcDREB148* were not expressed under low-temperature stress but were specifically expressed under LT-5-ALA treatment, indicating that these DEGs may play key regulatory roles in low-temperature stress and can be used as key candidate genes for low-temperature tolerance. In particular, *AcDREB1*, *AcDREB19*, *AcDREB46*, *AcDREB69*, *AcDREB117*, *AcDREB139*, and *AcDREB158* were enriched in the functional entries of the ethylene signal response, whereas *AcDREB69* was enriched in the functional entries of the ethylene signal response, oxygenate response, and endogenous stimulus-response, and *AcDREB148* functional entries were enriched for metabolic processes. More interestingly, compared with LT, when 5-ALA treatment was followed by low-temperature stress, the AcDREB genes, whose expression was upregulated, were downregulated. However, the AcDREB genes, whose expression was upregulated, were still upregulated. For example, compared with other genes, genes directly related to signaling pathways activated by abscisic acid were not enriched. Compared with those in the CK group, the *AcDREB19*, *AcDREB26*, *AcDREB37*, *AcDREB95*, *AcDREB139*, and *AcDREB152* genes were upregulated in the LT and LT-5-ALA groups. However, compared with those in the LT group, *AcDREB19*, *AcDREB37*, *AcDREB53*, *AcDREB95*, *AcDREB139*, and *AcDREB152* tended to be downregulated in the LT-5-ALA group, whereas *AcDREB26*, *AcDREB46*, *AcDREB69*, and *AcDREB148* were upregulated in the LT-5-ALA group, indicating that exogenous 5-ALA can relieve the damage caused by low-temperature stress by regulating the upregulation or downregulation of these 11 AcDREB genes.

However, KEGG enrichment analysis revealed that among the 46 DEGs, only 3 genes (*AcDREB69*, *AcDREB92*, and *AcDREB148*) were enriched in the plant hormone signaling and MAPK signaling–plant and phytopathogen interaction pathways. Among the three genes, only *AcDREB69* was related to the response to the ethylene signal (Figure 11A). Under normal growth conditions, the ethylene receptor ETR is active in the absence of ethylene [39]. However, low-temperature stress promotes an increase in the ethylene content, resulting in the degradation of its receptor and CTR1. By activating proprotein kinase 6 (MPK6), the accumulation of the EIN3 protein at low temperatures is promoted in EIN2-dependent and EIN2-independent manners, thereby significantly downregulating the expression of the direct target genes of EIN3 and *AcDREB69* and reducing its involvement in the response to low-temperature stress. The inability to regulate the expression of downstream ethylene response proteins and the promotion of the accumulation of ROS affect the cold tolerance of plants (Figure 11A). However, after 5-ALA treatment and then low-temperature stress, the reduction in ethylene content may restore the activity of the receptor ETR, relieve the degradation of the downstream EIN3 protein, and promote the upregulated expression of the target genes *AcDREB69*, *AcDREB92*, and *AcDREB148*, which regulate the transcription of downstream genes. Thus, the damage to kiwifruit seedlings caused by low-temperature stress was alleviated (Figure 11A). In addition, low-temperature stress leads to a decrease in the 5-ALA content in cells and an increase in the chlorophyll content in green tissues and promotes ROS accumulation [40] (Figure 11A). In addition, under low-temperature stress, the enzyme activities of DHAR and MFHAR decreased, which changed the redox state of the plants. The decrease in APX activity may have occurred because low temperatures hinder the stability or expression regulation of APX, limiting the regeneration ability of ascorbic acid (ASA). However, the increase in GR activity may be an adaptive response of plants to oxidative stress, which enhances their antioxidant ability through the reduced state of GSH, thereby affecting the efficiency of plant scavenging of hydrogen peroxide (H_2_O_2_) and indirectly regulating the low-temperature stress response (Figure 11A). Cold stress was followed by 5-ALA treatment to further increase the accumulation of enzymes and the contents of DHAR, MDHAR, GR, ASA, GAH, and GSSH [41,42], thereby increasing the efficiency of H_2_O_2_ removal in plants. The endogenous 5-ALA content can be directly increased via the exogenous application of 5-ALA [42], as can the chlorophyll content [43]. In addition, the enzyme activity and substance contents in the AsA–GSH circulatory system increased. However, the contents of hydrogen peroxide (H_2_O_2_) and superoxide (O^2•−^) decrease to remove massive amounts of accumulated ROS [44], thereby protecting cells from oxidative damage and improving the low-temperature tolerance of kiwifruit seedlings. Interestingly, the expression level of *AcDREB69* was significantly greater than that under low-temperature stress (Figure 11B), indicating that *AcDREB69* has a significant regulatory effect on low-temperature stress. The expression levels of the *AcDREB92* and *AcDREB148* genes related to the defense signal Pti6d increased under low-temperature stress, but the expression levels of the *AcDREB92* and *AcDREB148* genes further increased under LT-5-ALA treatment (Figure 11B), indicating that exogenous 5-ALA promotes not only the development of β-β-protection but also the promotion of LT-5-ALA treatment (Figure 11B). The response of plants to ethylene signaling also improves their cold tolerance by promoting the expression of defense-related genes.

To further analyze the functional structure of the genes *AcDREB69*, *AcDREB92*, and *AcDREB148*, which are involved in specific expression regulation, they were phylogenetically analyzed with known cold resistance genes via the maximum likelihood method. The results revealed that *AcDREB92* and *AcDREB148* clustered on the same branch of the phylogeny, and the nearest genes were *AT5G05410.1* and *XP002273838.1*, which contained the same conserved motifs as these two cold resistance genes did, suggesting that they may have the same function (Figure 12A). *AcDREB69* became an independent branch in the phylogeny and has no homologous genes. However, its conserved motifs are similar to those of the validated cold resistance genes, suggesting that it may have similar functions (Figure 12A). In particular, the conserved motifs 1, 2, and 5, which were present in 87% of the protein sequences, were highly conserved. However, the protein sequence of *AcDREB92* contained motif 6, in contrast to the remaining four genes, which had no conserved motif 6 (Figure 12A). This may be a result of the differences that exist between gene family genes under specific conditions. Therefore, the protein 3D structures of the above five genes were analyzed. The 3D structures also revealed some similarities between the genes that were closer in the phylogenetic tree (Figure 12C). In addition, protein interaction network analysis indicated that *AcDREB92*, *AcDREB148*, *XP002273838.1*, and *AT5G05410.1* formed reciprocal relationships and that *AcDREB69* did not form reciprocal relationships (Figure 12B). These findings suggest that *AcDREB92* and *AcDREB148* may have functions similar to those of the genes *XP002273838.1* and *AT5G05410.1*. Further investigation of the function of the DREB gene family involved in the regulation of exogenous 5-ALA to alleviate the response to low-temperature stress is crucial.

### 2.8. RT–qPCR Expression Analysis

To further verify the relative expression of the transcriptome data, we screened 15 DEGs (including internal controls) of the DREB genes that participated in the enrichment of GO and KEGG and used RT-qPCR to detect the relative expression levels. The results are shown in Figure 13A–N. The RT-qPCR expression of the genes followed the same trend as that of the transcriptome sequencing, except for the genes *AcDREB41*, *AcDREB45*, *AcDREB69*, and *AcDREB99*. The discrepancy in *AcDREB41*, *AcDREB45*, *AcDREB69*, and *AcDREB99* could be due to the temporal and spatial differences in the samples and indirectly indicate the presence of unstable expression of the gene, perhaps a pseudogene. However, many genes involved in the regulation of expression in roots irrigated with 3 g/L 5-ALA solution before low-temperature stress occurred. Compared with those in the CK treatment, under the LT treatment, with the exceptions of *AcDREB45*, *AcDREB53*, and *AcDREB69*, most genes tended to be upregulated. However, under LT-5-ALA treatment, in addition to the *AcDREB53* gene, the other two genes were downregulated. The expression of genes whose expression was significantly improved was upregulated, which was markedly different from that in LT. However, the differences between the *AcDREB7*, *AcDREB26*, and *AcDREB99* genes in the LT and LT-5-ALA treatments were not significant. However, the expression of the remaining two genes, in addition to the *AcDREB99* gene, was also upregulated to some extent, suggesting that rooting with the 5-ALA solution could increase the expression of these genes and increase the ability of plants to resist low temperatures. In addition, compared with those in the CK treatment, the genes *AcDREB19*, *AcDREB37*, *AcDREB53*, *AcDREB95*, *AcDREB139*, and *AcDREB152* were significantly upregulated under the LT treatment. However, root irrigation with 5-ALA solution relieved this upregulation and enhanced plant resistance to low-temperature stress. The genes *AcDREB7*, *AcDREB19*, *AcDREB26*, *AcDREB45*, *AcDREB53*, *AcDREB69*, *AcDREB95*, *AcDREB99*, *AcDREB139*, *AcDREB148*, *AcDREB152*, and *AcDREB184* were upregulated or downregulated, indicating that they play crucial regulatory roles in the response of exogenous 5-ALA to low-temperature stress. A comparison of the RNA-seq and qRT-PCR data of the above 14 genes in the three transcriptomes revealed that the expression trend was essentially consistent with the RNA-seq analysis results. Fold changes in the relative expression levels of the two genes in the three transcriptomes. The correlation coefficients (R^2^ values) were greater than 0.7 (Figure 13O–Q). The above results validated the accuracy of the expression levels of kiwifruit DREB-related genes and indicated that the above RNA-seq data were accurate.

## 3. Discussions

In this study, the kiwifruit DREB gene family was identified with 193 family members, which were completely aligned with those of *Arabidopsis thaliana* (56) [47], *Capsicum annuum* L. (49) [48], jackfruit (*Artocarpus heterophyllus* L.) (93) [49], *Camptotheca acuminata* (61) [50], and alfalfa (*Medicago sativa*) (172) [51]. Compared with other species, kiwifruit has more DREB. Cotton (*Gossypium barbadense*) (193) has the same number of DREB genes [52]. These results suggest that the number of DREB gene family genes is species dependent. Second, this study removed the sequences of the incomplete domain. That is, different analysis methods differ. The existence of several additional DREB genes cannot be excluded, but this makes the identification results more reliable. In kiwifruit, DREB significantly differed in protein size, instability coefficient, and isoelectric point, which was also reflected in a study of rapeseed [53]. In addition, the overall mean value was negative, possibly because the proteins have more polar amino acid residues or because their locations are conducive to the interaction of water molecules, resulting in higher hydrophilicity. Most members of this gene family are in the nucleus. However, a few are also located in the cytoplasm, indicating that AcDREB members function mainly in the nucleus. According to previous studies, the evolution of the genome and genetic systems requires the occurrence of gene duplication, which is a mode of gene amplification and one of the prime molecular mechanisms by which genes evolve to produce genes with new functions [54]. The asymmetric arrangement of genes on chromosomes has been reported to reveal information about their evolution [55]. Chromosome mapping and collinearity analysis of the kiwifruit DREB gene family revealed that the 193 kiwifruit DREB genes were randomly distributed across 29 chromosomes (e.g., Chr4, Chr7, and Chr24 were more concentrated). There was no clear correlation between chromosome length and gene number (e.g., the longest Chr1 contained only eight genes), suggesting that the distribution of genes may be driven by localized chromosomal structural or functional selection pressures. Collinearity analysis revealed a total of 183 pairs of segmental duplication genes, with Chr4 and Chr7 contributing the most, suggesting that segmental duplication is a central mechanism for the expansion of this gene family. In addition, a direct pairwise comparison of species revealed a closer evolutionary relationship between kiwifruit and apples.

Classification is an important part of functional analysis. A phylogenetic tree was constructed by multiple sequence alignment of DREB gene family members from kiwifruit (193) and *Arabidopsis thaliana* (56) [56]. The DREB genes of the two species are classified according to the DREB genes of *Arabidopsis*. The results revealed that kiwifruit and *Arabidopsis* were divided into six subfamilies. These findings illustrate that the evolutionary methods used for kiwifruit and *Arabidopsis* are relatively similar and do not remarkably differ. In this study, AcDREB was subdivided into six subgroups (A1–A6) based on the classification of *A. thaliana*, with the largest number of members in subgroup A4 and the smallest number of members in subgroup A3, a result that is consistent with that of banana [57], *oilseed rape* [53], and cotton [52]. When plants are under stress, the transcription and regulation of resistance genes must be initiated by promoters [58]. The six subgroups were further analyzed in the present study and were found to contain cis-acting elements associated with abiotic and biotic stresses as well as hormones. However, owing to functional differences, the number of cis-elements contained in each subfamily also differs. For example, the promoter regions of genes of subfamily A3 do not have a GC-motif, LTR, MBS, GATA-motif, MRE, MBSI, TATC-box, circadian, GCN4-motif, or O2-site cis-acting elements. However, almost all subfamilies contained relevant cis-acting elements related to abiotic and biotic stresses and plant hormone responses, indicating that the function of AcDREB may play a role in plant hormone responses and abiotic stress responses, which is similar to the results of MnDREB in *mulberry* [59]. The DREB family is a subfamily of the ethylene response factor (ERF) gene family, which contains a conserved AP2 domain consisting of 57–70 amino acid residues that bind to the dehydration response element/C repeat sequence in the promoter region of stress-inducible gene (DRE/CRT) interactions to regulate the expression of genes under various downstream stresses. The ERE also plays an important regulatory role and endows plants with stress tolerance [15,21]. In this study, almost all the DEGs of the kiwifruit DREB gene family contained ERE cis-acting elements; 13 DEGs contained DRE cis-acting elements, and 20 DEGs contained LTR cis-acting elements. Among the DEGs, *AcDREB69*, *AcDREB92*, and *AcDREB148* were involved in the regulation of ERE cis-acting elements, whereas *AcDREB69* and *AcDREB148* were involved in the regulation of LTR cis-acting elements. Interestingly, DRE acts as a dehydration response element, and the *AcDREB69*, *AcDREB92*, and *AcDREB148* genes are not involved. These results suggest that ERE and LTR cis-acting elements are key cis-acting elements involved in the cold resistance of kiwifruit. Mizoi et al. reported that DREB transcription factors could regulate the expression of antistress genes through ABA-dependent and ABA-independent pathways [60]. In tomatoes, ectopic expression of *SlDREB3* can increase the growth of plant roots and improve the photosynthetic rate by reducing ABA levels [61]. Taken together, the cis-acting elements of the AcDREB subgroup of genes may also regulate physiological levels and response to adverse stress in kiwifruit by binding to some target genes. Moreover, the GO functional annotation results revealed that the AcDREB gene family was enriched in a greater number of genes in the functional entries of organic matter metabolism, environmental stress response (low temperature, drought, and temperature), and hormone signaling pathways (ethylene, abscisic acid, and jasmonic acid), e.g., *AcDREB92* and *AcDREB148* were enriched in metabolic processes in the GO function, and *AcDREB69* in the endogenous stimuli, signaling, ethylene, metabolic processes, and oxygenates were enriched. In the KEGG enrichment analysis, only 36 genes were enriched in plant hormone signal transduction, the MAPK signaling pathway, plant–pathogen interactions, and the zygosomal pathway, and genes from different pathways presented subgroup-specific distributions (such as A-4, which is enriched in signaling pathways; MAPK signaling; and subgroups A-4, A-5, and A-6, which are collectively involved in pathogen interactions), such as the *AcDREB69* gene, which is enriched in the ethylene pathway of phytohormone signaling, and the MAPK signaling–plant pathway in the ethylene pathway. The *AcDREB92* and *AcDREB148* genes were enriched in the plant–pathogen interaction pathway, and these three key genes were enriched in the location of the cis-acting element analysis; similarly, these genes are involved in the regulation of the cis-acting ERE. This result may be related to the number of genes in the A-2 and A-4 subgroups. These findings further reveal that, under low-temperature stress, the AcDREB gene family responds to adverse environments by regulating the expression levels of hormones and stimulating plant defense responses to regulate plant growth and development. This result was similar to that of the homeopathic element. These results indicate that the kiwifruit DREB gene family plays crucial roles in hormone signal transduction, abiotic stress responses, and plant growth.

When plants are subjected to adverse stress, they respond through corresponding life activities, maintaining normal intracellular activities. Exogenous 5-ALA has a function similar to that of plant growth-regulating substances and significantly promotes plant stress tolerance, growth, and development [29]. Previous studies have shown that 5-ALA administration can increase the activity of antioxidant enzymes and reduce damage to cell membranes caused by ROS. For example, the application of 5-ALA enhances the cold tolerance of common bean seedlings by regulating hormone signal transduction, chlorophyll metabolism, and photosynthesis under low-temperature stress [62]. The application of 5-ALA can effectively protect soybean plants from low temperatures by increasing the activity of heme proteins (such as catalase (CAT)) and promoting heme catabolism, resulting in the production of high levels of the antioxidants biliverdin and carbon monoxide. The destructive effect of stress [63] 5-ALA application increases the seed respiratory rate and ATP synthesis of GSH, total glutathione, and ascorbic acid concentrations of AsA, as well as the activities of SOD, CAT, APX, and GR, while decreasing the contents of MDA and H_2_O_2_ and the contents of O^2•–^ release and increasing H+-ATPase activity and the endogenous 5-ALA concentration, thus effectively protecting *E. nutans* seeds from cold-induced oxidative damage during germination [42]. In this study, the 5-ALA content of kiwifruit seedlings decreased significantly under low-temperature stress. However, the application of exogenous 5-ALA increased the 5-ALA content, and it is possible that exogenous 5-ALA not only directly supplements the deficiency of 5-ALA in plants under low-temperature stress but also promotes the synthesis or accumulation of endogenous 5-ALA synthesis or accumulation. Under low-temperature stress (LT), the Chl a and Chl b contents of kiwifruit seedlings were greater than those under CK. After the roots were irrigated with exogenous 5-ALA, the Chla and Chlb contents significantly increased, similar to the results of Korkma, Anwar, and Xu [64,65,66]. The application of exogenous 5-ALA relieved the inhibitory effect of low-temperature stress on the activities of DHAR and MDHAR to some extent, and the DHAR and MDHAR activities in the LT-5-ALA treatment group were slightly greater than those in the LT treatment group. In addition, the GR activity in the LT-5-ALA treatment group was significantly greater than that in the LT treatment group (18.58% increase) and the CK group (72.27% increase). The increase in DHAR, MDHAR, and GR activity after root irrigation with exogenous 5-ALA was consistent with previous findings. The results of human studies have been consistent [67,68,69]. Low-temperature stress significantly induced the accumulation of antioxidants (AsA, GSH, and GSSG) in the kiwifruit seedlings. Compared with the CK group, the LT-treated group presented significantly increased AsA, GSH, and GSSG contents, which were similar to the results of previous studies [70], indicating that kiwifruit seedlings respond to oxidative damage under low-temperature stress by increasing their antioxidant levels. After the roots were irrigated with exogenous 5-ALA, the antioxidant contents further increased; the AsA, GSH, and GSSG contents of the LT-5-ALA treatment group increased by 19.233%, 6.031%, and 43.430%, respectively, compared with those of the LT treatment group. The increase in the contents of antioxidant substances after ALA was related to that reported by Xinhui Luo [71]. The study results were consistent with previous findings. In addition, numerous studies have shown that gene expression and hormone signaling are involved in the response to low-temperature stress [72,73,74,75,76,77]. In particular, the DREB gene plays a crucial role in coping with abiotic stresses [14,78,79,80]. For example, the overexpression of MbDREB1 in *Arabidopsis* revealed that the expression of both the ABA-dependent stress-inducible gene rd29B and the ABA-independent COR gene was activated, ultimately increasing the cold tolerance of the plants [81]. VvDREB2A improves plant resistance to cold stress by scavenging reactive oxygen species, increasing the amount of RFO, and increasing the expression levels of cold stress-related genes [82]. Transgenic *Arabidopsis* plants overexpressing MaDREB14/22/51 presented increased cold tolerance through reduced MDA content and increased PRO and soluble sugar contents [57]. The overexpression of OsDREB1A in transgenic plants confers resistance to cold stress [83]. These results suggest that gene expression regulation may be involved in the corresponding response pathways. In this study, the expression levels of *AcDREB69* and *AcDREB148* significantly increased after LT-5-ALA root irrigation. However, the expression level of *AcDREB92* was significantly upregulated both under low-temperature stress and in roots irrigated with 5-ALA under low-temperature stress, and the expression pattern was consistent with the results of the functional analysis. In this study, low-temperature stress promoted an increase in ethylene content, which led to the degradation of its receptor and CTR1. By activating proprotein kinase 6 (MPK6), the accumulation of EIN3 protein at low temperatures is promoted in EIN2-dependent and EIN2-independent manners, thereby significantly downregulating the expression of the direct target genes of EIN3 and *AcDREB69* and reducing its involvement in the response to low-temperature stress. The inability to regulate the expression of downstream ethylene response proteins and promote the accumulation of ROS affects the cold tolerance of plants (Figure 11A). These results suggest that ethylene signaling plays a vital regulatory role in the low-temperature stress responses. For example, in grapes, the expression of genes related to the synthesis and signaling pathways of ETH, IAA, and ABA is upregulated under low-temperature stress; ETH (ethylene) increases significantly under low-temperature stress and positively regulates cold tolerance [84]. In addition, low-temperature stress can promote the accumulation of EIN2-dependent EIN3/EIL1 proteins, which directly regulate the expression of ERFs and accelerate the removal of ROS in plants, thereby increasing cold tolerance in plants [46]. However, no studies have reported that the DREB gene is involved in low-temperature stress regulation by exogenous 5-ALA. This study revealed that low-temperature stress followed by 5-ALA treatment also affected the ethylene content, which may restore the receptor ETR and promote the upregulated expression of the target genes *AcDREB69*, *AcDREB92*, and *AcDREB148*, thereby alleviating the effects of low-temperature stress on kiwifruits. caused harm. These results suggest that the DREB gene family is involved in plant cold tolerance regulation mediated by exogenous 5-ALA.

The type and number of motifs in a transcription factor determine its function [85]. Genes with relatively high homology are highly similar in structure and function [52,86,87]. *The Xanthoceras sorbifolium* phylogeny belongs to the same subgroup. The XsDREB and *Arabidopsis* genes have highly analogous gene structures and conserved motif compositions [88]. In this study, we found that genes in the phylogenetic tree that are close together or in the same branch show some similarity in their three-dimensional structures in addition to containing similar conserved motifs (Figure 12). These results suggest that *AcDREB92* and *AcDREB148* may have functions identical to those of genes that are closer to each other on the developmental tree. These findings are crucial for further studies on the function of the DREB gene family in the participation of 5-ALA in the regulation of the response to low-temperature stress.

## 4. Materials and Methods

The plant material used in this study was ‘Hongyang’ kiwifruit, which was obtained from Northwest Agriculture and Forestry University and cultivated by the research group. The authors harvested young leaves and rinsed them well under running water for the cultivation of live kiwifruit seedlings. First, based on kiwifruit genomic data, the DREB gene family in kiwifruit was analyzed via bioinformatics methods. Next, the physiological indicators of kiwifruit in response to low-temperature stress were measured, and the AcDREB genes whose expression significantly differed were obtained through transcriptome sequencing and real-time PCR analysis. Finally, based on the above analysis results, the key AcDREB and cold tolerance genes were screened for comparative analysis. The URLs of the databases and software used in this study are listed in Appendix A.

### 4.1. Data Acquisition

The genome of the kiwifruit cultivar ‘Hongyang’ (*Actinidia chinensis* Hong Yang v3) was downloaded from the kiwifruit genome database [89,90] (20 November 2024), the rice genome data were obtained from the Rice Genome Database [91] (20 November 2024), and the gene sequence file of the apple GFF file was downloaded from the GDR Rosaceae gene database (20 November 2024). The *Arabidopsis* genome data were obtained from the Ensembal Plants Database (20 November 2024), and the sequence information of the *Arabidopsis* DREB family was downloaded from the online TAIR database [47] (20 November 2024). In addition, the protein sequences of *soybean GmDREB3* (GenBank accession no. DQ208969) [27] (27 April 2025), apple ‘*Dwarf apple*’ *MbDREB1* (GenBank accession number: EF582842) [81] (27 April 2025), and tea tree *CsCBF1* (GenBank accession number: EU563238) [92] (27 April 2025) were obtained from the NCBI GenBank database. The protein sequence of the grape ‘Pinot Noir’ cultivar *VvDREB2A* (*XP_002273838.1*) was obtained from a previous study [82] (27 April 2025). The protein sequence of the *alfalfa cultivar* Xinjiangdaye was downloaded from the figshare data repository [51] (27 April 2025).

### 4.2. Identification and Classification of the AcDREB Gene Family

The sequence of the AP2/ERF conserved domain (PF00847) was downloaded from the protein family analysis and modeling (Pfam) website and used to further identify AP2/ERF family members with e-values < e^−5^ [49]. An online simple modular architecture research tool [93] was used to confirm the conserved domain of DREB, and a hidden Markov model was used [94] (9 December 2024). Biosequence analysis was performed. The standard of phylogenetic tree construction in MEGA11 software was set to the maximum likelihood method of paired deletion, the Poisson replacement model was used, the bootstrap was set to 1000 repeats [95] (9 December 2024), and Chiplot online software was used to visualize the evolutionary tree [96] (10 December 2024). Finally, based on the grouping of *Arabidopsis* DREB proteins, the 193 kiwifruit DREB proteins were renamed and grouped, and the naming method was as follows: AcDREB + serial number. The DREB proteins from kiwifruit and *Arabidopsis* were compared and divided into groups (A-1, A-2, A-3, A-4, A-5, and A-6). Amino acids, theoretical isoelectric points, and molecular weights were analyzed via ExPASy ProtParam (13 December 2024). The instability index, aliphatic index, and grand mean hydration value were analyzed via the Protein Parameter Calc program in TBtools [97]. Subcellular localization was predicted via WoLF PSORT (13 December 2024).

### 4.3. Gene Structure and Conserved Motif Analysis of the AcDREB Gene Family

Gene Structure Display Server 2.0 was used [98] (20 December 2024), and the DREB family and the full-length sequences of the kiwifruit genome files were employed to construct the gene structure. With the default parameters, Multiple Em for Motif Elicitation (MEME) was used [87] (as of 9 December 2024) online software (v5.5.7) was used to analyze the conserved motifs of the DREB family to analyze differences among the AcDREB family members. The above results were visualized using TBtools software [99].

### 4.4. Chromosomal Positioning and Collinearity Analysis

Chromosomal location information for the DREB family was extracted from the kiwifruit genome files and analyzed via TBtools (v2.031) [100], which is represented graphically. Gene location visualization was performed via the GTF/GFF program. The collinear gene pairs of the AcDREB genes were analyzed using one-step MCScanX; the BLASTP results and gene location information were used as inputs for TBtools [101]. The proximal scatter tandems of the AcREB family genes were identified using TBtools and visualized via TBtools Advanced Circos [86]. Interspecific collinearity was also constructed via TBtools software.

### 4.5. Analysis of Cis-Acting Elements in Kiwifruit DREB Subgroups

Based on the clustering results of the gene IDs and phylogenetic tree obtained from identification, the upstream 2000 bp nucleotide sequence of each gene was extracted from the genome annotation files (GFF3) via TBtools software (v2.031), and the online analysis software PlantCARE was used for promoter homeomorphic element analysis [53] (11 December 2024). The sequence was finally visualized via TBtools Basic Biosequence View.

### 4.6. GO and KEGG Enrichment Analyses of the Kiwifruit DREB Gene

The protein sequences of 193 DREB genes in kiwifruit were submitted to the EggNOG-MAPPER database (21 December 2024) for gene function annotation. The obtained results were subsequently subjected to GO function and KEGG pathway enrichment analysis via TBtools. Graphing and visualization were performed via the online tool ChiPlot (25 December 2024).

### 4.7. Analysis of the Expression Pattern of the AcDREB Gene in Kiwifruit Under Low-Temperature Stress Regulated by Exogenous 5-ALA

#### 4.7.1. Plant Materials and Experimental Design

Young leaves of ‘Hongyang’ kiwifruit were collected from September 2023 to April 2024 at the Key Laboratory of Forest Resources Conservation and Utilization of Southwest Forestry University (longitude: 102.755543; latitude: 25.05731) to conduct tissue culture experiments. The complete ‘Hongyang’ kiwifruit seedlings were used as test materials. Briefly, the leaves were rinsed with tap water for 12 h, disinfected with sodium hypochlorite (NaOCl) on an ultraclean workbench, and then rinsed 3 to 5 times with pure water. The plants were placed in tissue culture flasks for callus induction, adventitious bud induction, and rooting induction. During the callus culture period, the tissue culture flasks were cultured in the dark at 26 °C for approximately 15 days. The latter induction was performed at 26 °C under a light intensity of 2000 lx and under 14 h of light conditions every day.

When the plants reached a height of 10 cm, uniformly sized seedlings were selected and divided into three groups: the control group (CK), the low-temperature treatment group (LT), and the low-temperature + exogenous element group (LT-5-ALA). Through the preliminary screening of exogenous elements to relieve low-temperature stress (−2 °C for five hours), the most suitable concentration of exogenous elements was 3 g/L 5-ALA, and 6 mL of diluted exogenous elements was injected into each bottle of the medium. The control group and the low-temperature group were injected with the same volume of distilled water, and the low-temperature + exogenous element (LT-5-ALA) group was injected with 3 g/L 5-ALA. The kiwifruit seedlings with complete plants were transferred to culture bottles filled with distilled water and 5-ALA. Seven days of treatment were performed. After seven days, the seedlings in each treatment group were further divided into three groups. The seedlings in the CK group were maintained under normal conditions, whereas the seedlings in the other two groups were grown under low-temperature stress at −2 °C. Notably, there were three treatment groups: (1) CK: seedlings that were injected with 6 mL of distilled water for 7 days in the tissue culture flasks with medium and grown under normal conditions; (2) LT: seedlings grown under normal conditions; seedlings grown under low-temperature stress were injected into the tissue culture flasks with 6 mL of distilled water for seven days; and (3) LT-5-ALA: 6 mL of 5-ALA was injected into the tissue culture flasks with medium for seven days and then added to the seedlings grown under low-temperature stress. A completely randomized block design was used for the four treatments.

Physiological indicators were measured, and transcriptomic experiments were performed. Three biological replicates were performed for each treatment. In brief, leaf samples were collected after five hours of low-temperature treatment for physiological experiments and transcriptome sequencing. All the samples were immediately frozen in liquid nitrogen and stored in a −80 °C freezer until subsequent analysis.

#### 4.7.2. Measurement of Physiological Indicators

##### Determination of the 5-ALA Content and Chlorophyll Content

To determine the 5-ALA content in the leaves, a double-antibody one-step sandwich enzyme-linked immunosorbent assay (ELISA) was performed. To the coated microwells precoated with the 5-ALA antibody, the sample and detection antibody were added sequentially according to the kit instructions, incubated, and washed thoroughly. The substrate TMB was used to develop color. TMB was converted to blue under POD catalysis and finally turned yellow under acidic conditions. The shade of color was positively correlated with the 5-ALA content of the sample. The absorbance (OD value) was measured at a wavelength of 450 nm via a microplate reader to calculate the sample content.

The chlorophyll content was determined via a plant chlorophyll (Chl) content kit. Approximately 0.1 g of sample was weighed, 1 mL of extraction buffer and a small amount of reagent I (approximately 50 mg) were added, and the mixture was ground thoroughly in a −40 °C tissue grinder under dark or light conditions. After washing with extraction solution, all the rinse solution was transferred to a 10 mL centrifuge tube, and the volume was diluted to 10 mL with extraction solution. The samples were placed under the above conditions or wrapped and extracted with tin foil for three hours (during this period, the samples were mixed by inversion twice). The tissue residue at the bottom turned completely white, indicating that the extraction was complete. If the tissue residue does not completely turn white, leaching continues until it turns white completely. Two hundred microliters of extraction solution and two hundred microliters of extraction buffer were collected in a 96-well plate and recorded as the measurement tube and the blank tube. The absorbance value A of the chlorophyll extract was read at 665 nm and 649 nm: ΔA_665_ = (A measurement − A blank) _665_; ΔA_649_ = (A measurement − A blank) _649_. The chlorophyll a content, chlorophyll b content, and total chlorophyll content in each sample were subsequently calculated via the Lambert–Beer law formula.

##### Determination of Antioxidant Enzyme Activities

Dehydroascorbate reductase (DHAR) (EC 1.8.5.1) and monodehydroascorbate reductase (MDHAR) (EC 1.6.5.4) activity analyses were carried out via the DHAR kit, where the supernatant and the test reagent were placed in sequence according to the steps in the DHAR and MDHAR kit assay manuals. The absorbance values were measured at 412 nm and 340 nm.

Glutathione reductase (GR) (EC 1.6.4.2) activity was analyzed via a GR kit. The enzyme marker was preheated for more than 30 min. The wavelength was adjusted to 340 nm for measurement according to the steps of the kit instructions.

Ascorbate peroxidase (APX) (EC 1.11.1.11) activity was analyzed via a BCA protein content assay kit. The absorbance values were determined at 10 s and 2 min after preheating the enzyme marker for more than 30 min and adjusting the wavelength to 290 nm according to the instructions of the kit.

##### Ascorbic Acid–Glutathione Cycle Analysis

The contents of reduced glutathione (GSH), oxidized glutathione (GSSG), and reduced ascorbic acid (AsA) were determined via detection kits. Approximately 0.1 g of tissue was weighed, and 1 mL of extraction solution was added for homogenization in an ice bath. After centrifugation at 12,000 rpm at 4 °C for 15 min, 100 μL of the supernatant was collected, and the GSH content was determined at an absorbance of 412 nm according to the manufacturer’s instructions.

A total of 0.1 g of sample was weighed, 1 mL of extraction solution was added, and the mixture was homogenized in an ice bath. After centrifugation at 8000× *g* for 10 min at 4 °C, the supernatant was placed on ice until assayed. The 2-nitro-5-mercaptobenzoic acid produced by the reaction between glutathione and DTNB has the characteristics of maximum light absorption at 412 nm. The original GSH in the sample was inhibited by 2-vinylpyridine, and then, glutathione was used to inhibit the original GSH in the sample. Peptide reductase reduces GSSG to GSH, and GSH and DTNB react to form a complex. The microplate reader was preheated for more than 30 min, the reagents were quickly mixed, and the absorbance of GSSG was measured at 412 nm for 30 s and 150 s.

The ASA content was detected via a reduced ascorbic acid (ASA)/vitamin C content detection kit. Approximately 0.1 g of tissue (approximately 0.5 g of sample with sufficient water) was weighed, and 1 mL of extraction solution was added for homogenization in an ice bath. After centrifugation at 4000 rpm for 10 min, the supernatant and reagents were mixed homogeneously. After incubation in a 30 °C water bath for 60 min, 200 μL was transferred to a 96-well plate, and the absorbance of each tube was read at 534 nm.

#### 4.7.3. RNA Extraction, cDNA Library Construction, and Sequencing

Total RNA from treated kiwifruit leaves was extracted via the Plant Total RNA Extraction Kit (Basilica Biotechnology, Inc., Seattle, WA, USA). The integrity of the RNA and the presence of DNA contamination were analyzed via 1% agarose gel electrophoresis. The concentration and purity of the RNA were detected with an ultramicro spectrophotometer. After being subjected to the test, the RNA samples from the kiwifruit leaves were sent to Basilica Biotechnology, Inc. (Seattle, WA, USA), for cDNA library construction and sequencing. The sequencing results were uploaded to the NCBI database under the accession number PRJNA1208675.

#### 4.7.4. Expression Pattern Analysis of the Kiwifruit DREB Gene Family

The reference genome of the transcriptome sequencing data was the whole kiwifruit genome, and the filtered reads were subjected to HISAT2 software (v2.0.4) and aligned with the reference genome. The HTSeq statistics were compared to the read count value for each gene, which was used as the original expression level of the gene. Calculation and average expression level (fragments per kilobase per million bases, FPKM) transformation were performed for each transcribed region. The paired-end reads from the same fragment were counted as a fragment, and the expected count data of the expression levels of genes and transcripts were obtained. Genes with FPKM values greater than zero were considered expressed genes. The Poisson distribution method (Poisson Dis) was used for differential gene detection on the expected count data. The *p*-value of the detection was corrected for multiple hypothesis testing [102] and by controlling the false discovery rate (FDR) [99] to determine the domain value of the *p*-value. Then, using the read count information, the R language package edgeR was employed for differentially expressed gene analysis. The screening thresholds for the final DEGs were FDR < 0.05, log_2_FC > 1, or log_2_FC < −1 [103].

#### 4.7.5. qRT-PCR Analysis

To further validate the relative expression of the transcriptome data, fourteen DEGs were randomly selected for qRT-PCR validation. The total RNA of all the samples was extracted via an RNA extraction kit, after which the RNA was reverse transcribed into cDNA via a Monad kit for qRT-PCR analysis. Primer 6.0 was used to design the primers, and the reaction procedure and PCR primer design are shown in Appendix A Actin was used as the internal reference gene, and standard qRT-PCR analysis was performed via a 2x SYBR Green PCR Master Mix (Universal) kit. Three biological replicates were set up for each of the above steps, and the relative expression of DEGs was calculated via the 2^−ΔΔCt^ method [104]. Statistical analysis was performed via SPSS (v24), and *p*-value were calculated via the independent samples *t* test [105].

### 4.8. Phylogenetic, Conserved Motif, Protein Structure, and Interaction Analyses of the Cold Resistance DREB Gene Family

At present, many genes of the DREB gene family have been shown to have cold resistance (Appendix A). Our previous studies revealed that transcription factor genes with high homology are highly similar in structure and function [103,106,107]. Therefore, comparison with cold-resistant DREB genes with known functions helps identify candidate homologous genes, predict their gene functions, and analyze their role in exogenous 5-ALA alleviation of low-temperature stress in kiwifruits. The screened genes of the key DREB gene family were analyzed with genes that were verified to regulate low-temperature stress. MEGA was used for sequence alignment, and phylogenetic tree analysis was performed via maximum likelihood (ML). Conserved motifs play a vital role in cellular processes by mediating protein–protein interactions [108]. Therefore, the conserved motifs were analyzed via the MEME online website (MEME) (27 April 2025) submission form. The three-dimensional structures of the corresponding proteins were analyzed via the SWISS-MODEL online website (28 April 2025). For protein structure interaction analysis, the STRING protein–protein interaction (PPI) database (29 April 2025) was used to construct network interaction relationships for the genes of the key DREB gene family.

## 5. Conclusions

This study identified 193 DREB genes from the kiwifruit genome. According to the clustering results of the evolutionary tree, the genes were divided into six subfamilies, A1 to A6. Homeopathic element analysis revealed that the kiwifruit DREB gene family plays important roles in hormone signal transduction, abiotic stress responses, and plant growth. Forty-six DREB genes related to the response of exogenous 5-ALA to low-temperature stress were differentially expressed in kiwifruit. Among them, under induction with 3 g/L 5-ALA, *AcDREB7*, *AcDREB26*, *AcDREB37*, *AcDREB45*, *AcDREB53*, *AcDREB69*, *AcDREB95*, *AcDREB139*, *AcDREB148*, and *AcDREB152* were involved in the response to cold stress. In particular, DEGs such as *AcDREB69*, *AcDREB92*, and *AcDREB148* were involved in the response to ethylene and defense signals. These genes participate in the regulatory effect of exogenous 5-ALA on low-temperature stress in kiwifruits by activating the transcription of downstream target genes. In addition, after root irrigation at 3 g/L, exogenous 5-ALA inhibited the accumulation of substances such as APX and DHAR and promoted an increase in chlorophyll. The contents of 5-ALA, enzymes, and substances such as MDHAR, GR, ASA, GAH, and GSSH were significantly reduced. The accumulation of ROS in kiwifruit further increased to accelerate the removal of ROS and increase the cold tolerance of the kiwifruit. Notably, *AcDREB69*, *AcDREB92*, and *AcDREB148* may be similarly cold tolerant to genes that are closer together in the developmental tree and could serve as key genes for cold tolerance. This study is the first to investigate the function of the AcDREB gene in low-temperature stress regulated by exogenous 5-ALA. These findings reveal the regulatory mechanism of DREB in the participation of exogenous 5-ALA in alleviating low-temperature stress. However, transgenic verification has not been carried out to accurately confirm the role of these genes under low-temperature stress; further experiments in this area can be strengthened in the future, and the detailed roles of the above genes under low-temperature stress have been verified.

## Figures and Tables

**Figure 1 plants-14-02560-f001:**
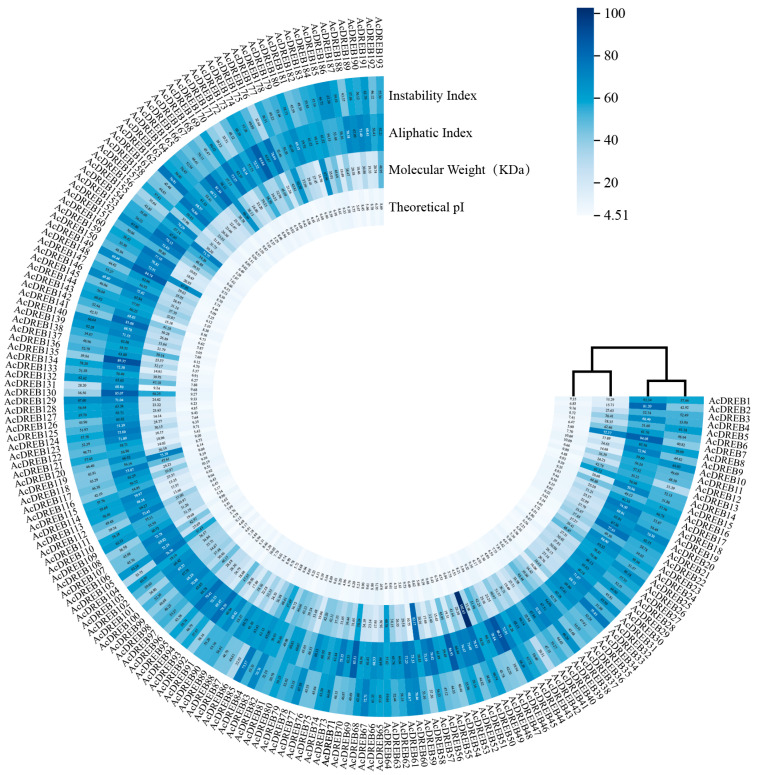
Analysis diagram of the physicochemical properties of the AcDREB gene. The numbers in the heatmap represent numerical values, and the larger the value is, the darker the color; the names of the genes shown in the outermost circle and the inner circle from inside to outside are the theoretical pI, molecular weight (kDa), aliphatic index, and instability index.

**Figure 2 plants-14-02560-f002:**
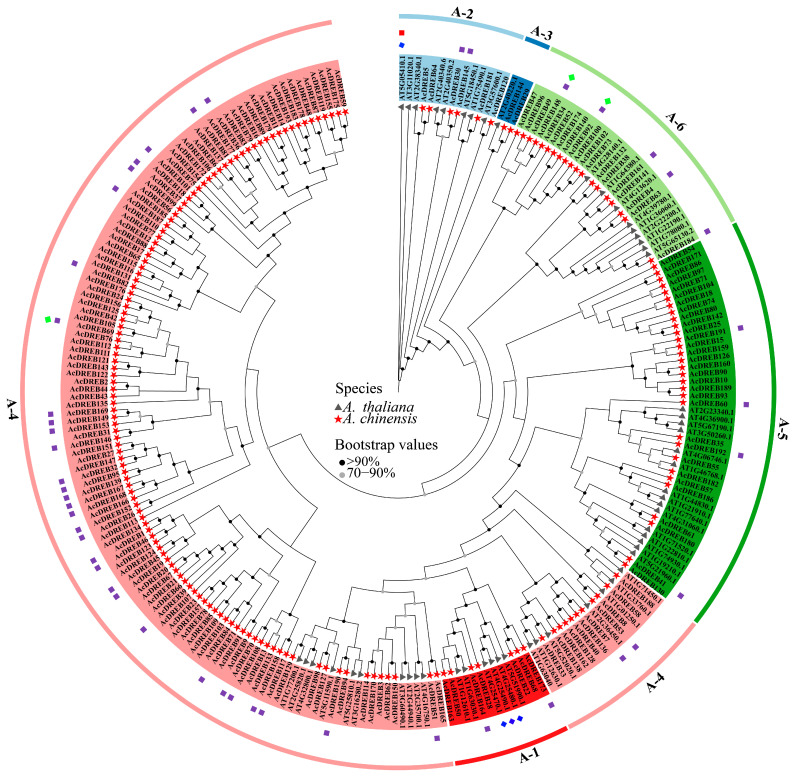
Phylogenetic trees of the DREB gene family in *Arabidopsis* and kiwifruit. The phylogenetic tree was constructed via the maximum likelihood (ML) method. A total of 1000 repetitive boot values were used. Different groupings are represented by different colors; the AcDREB gene is represented by a red five-pointed star, the AtDREB gene is represented by a gray triangle, and the purple square is represented. Differentially expressed genes are represented. The green diamonds represent genes in the KEGG pathway among the DEGs, the blue diamonds represent cold resistance genes in *Arabidopsis*, and the red squares represent drought resistance genes in *Arabidopsis*.

**Figure 3 plants-14-02560-f003:**
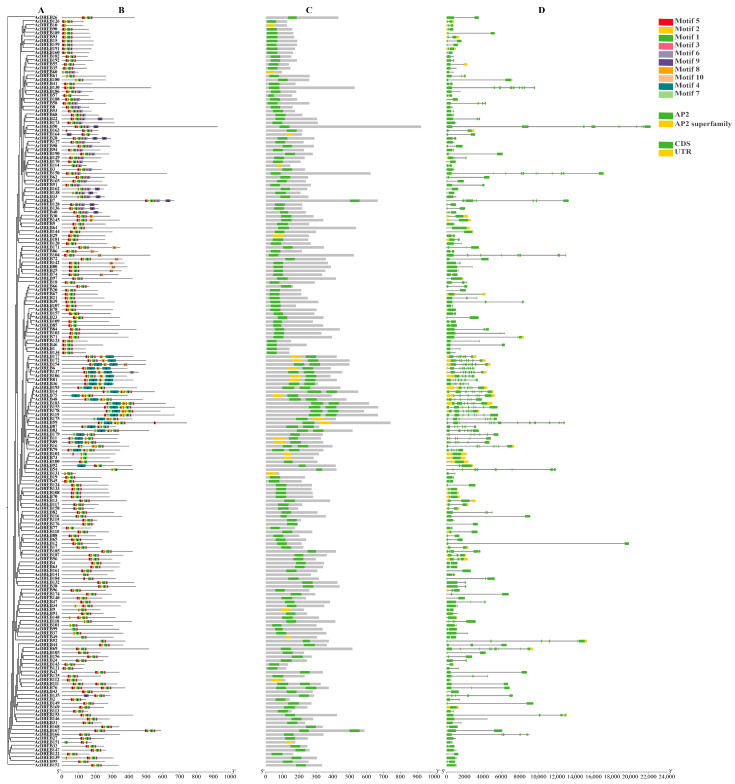
Diagram of the conserved motifs and gene structure of the AcDREB gene family. (**A**) Members and phylogenetic relationships of the DREB gene family in kiwifruit. (**B**) Distribution of the conserved motifs of AcDREB. The motifs are represented by colored boxes, and the black line represents the relative length of the proteins. (**C**) Gene structure of the AcDREB gene. (**D**) Exon–intron structure of the AcDREB gene. Green represents the CDS, yellow represents the UTR, and the line in the middle represents introns.

**Figure 4 plants-14-02560-f004:**
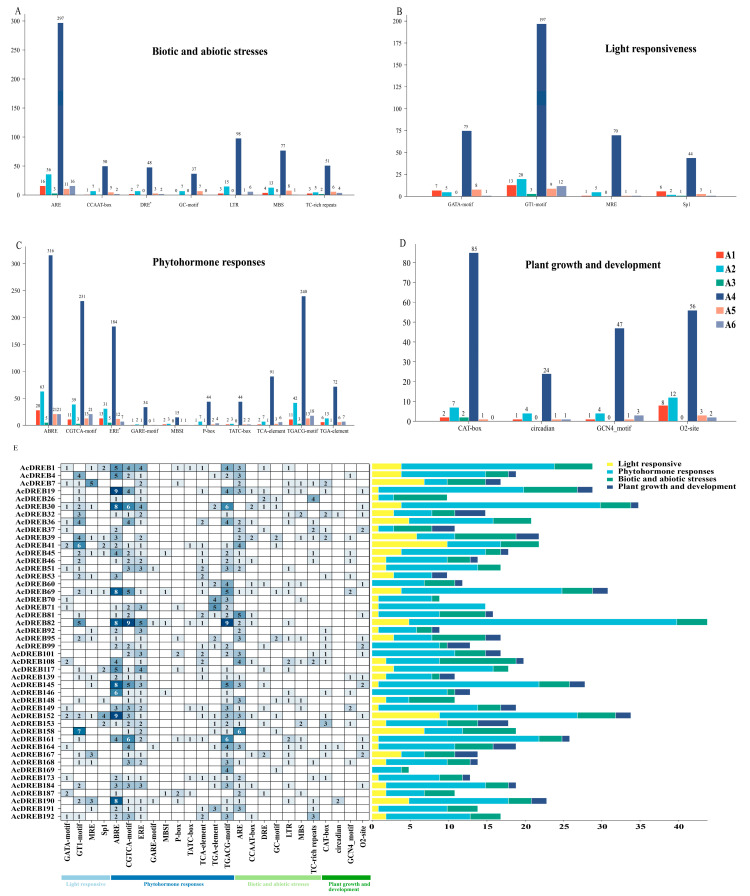
Summary of the DEGs of the kiwifruit DREB gene and the number of cis-acting elements for each subfamily in the four major classifications. The cis-elements contained in each subfamily are indicated in different colors. * Indicates cis-acting elements associated with low-temperature stress. (**A**) Summary of the number of cis-acting elements and the number of each subfamily involved in response to abiotic stress and biotic stress. (**B**) Summary of the number of cis-acting elements and the number of each subfamily included in the light response. (**C**) Summary of the role of hormones in each subfamily and the number of cis-acting elements involved in the reaction. (**D**) Summary and quantity of cis-acting elements in each subfamily involved in plant growth and development. (**E**) Distribution of cis-elements in the 2000 bp promoter region upstream of 46 DREB differentially expressed genes.

**Figure 5 plants-14-02560-f005:**
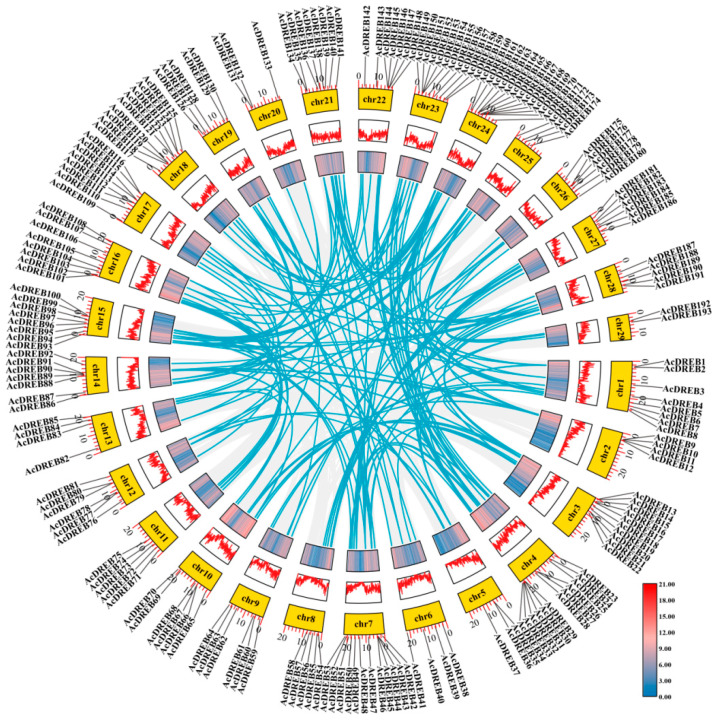
Chromosomal location map of the kiwifruit DREB gene. Chromosomal location map of the kiwifruit DREB gene. The gray line represents all the repetitive genes in the kiwifruit genome; the blue line represents the repetitive genes of the kiwifruit DREB fragment; and the orange pattern represents the kiwifruit chromosome.

**Figure 6 plants-14-02560-f006:**
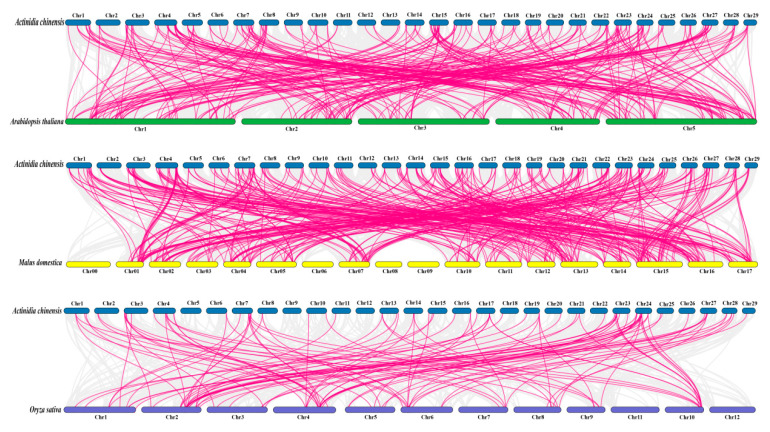
Collinearity plot among different species of kiwi, rice, *Arabidopsis*, and apple. The gray lines indicate duplicates of blocks, the pink line represents the collinear DREB gene pair, the blue line represents the kiwifruit chromosome, the green line represents the *Arabidopsis* chromosome, the yellow line represents the apple chromosome, and the purple line represents the rice chromosome.

**Figure 7 plants-14-02560-f007:**
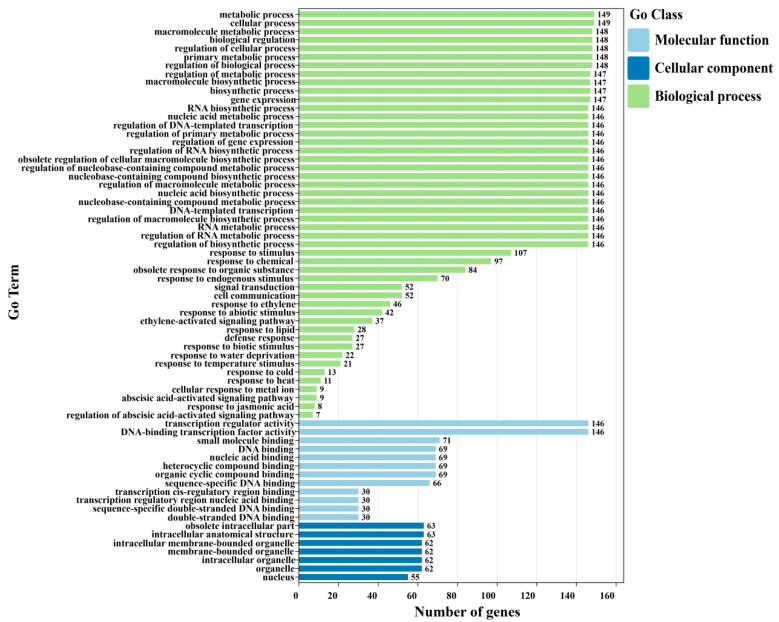
Annotation results of the GO-level 3 of the AcDREB gene. The molecular function represents a molecular function, the cellular component represents a cellular component, and the biological process represents a biological process, which is represented in light blue, dark blue, and green, respectively. The *x*-axis represents the number of gene functions, and the *y*-axis represents the annotated functions of genes.

**Figure 8 plants-14-02560-f008:**
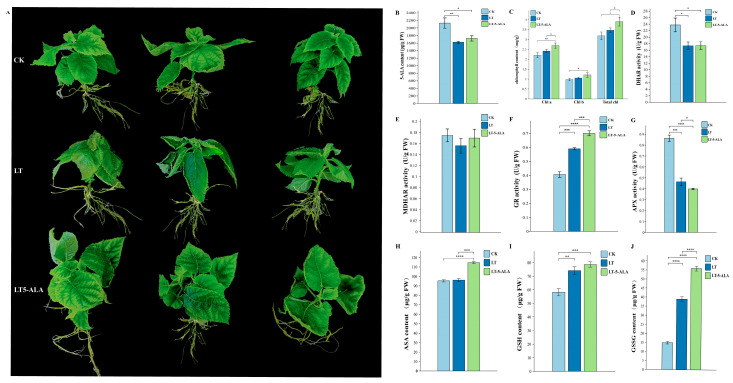
Effects of 5-ALA on the leaves of kiwifruit plants under low-temperature stress. (**A**) 5-ALA relieved the response of kiwifruit seedlings under low-temperature stress; (**B**) 5-ALA content; (**C**) Chl content; (**D**) DHAR activity; (**E**) MDHAR activity; (**F**) GR activity; (**G**) APX activity; (**H**) ASA activity; (**I**) GSH activity; (**J**) GSSH activity. The data are expressed as the means ± standard deviations of three biological replicates. The vertical line on the bar graph in the diagrams is the standard deviation line, and the * sign represents the difference. The more * signs there are, the more significant the difference.

**Figure 9 plants-14-02560-f009:**
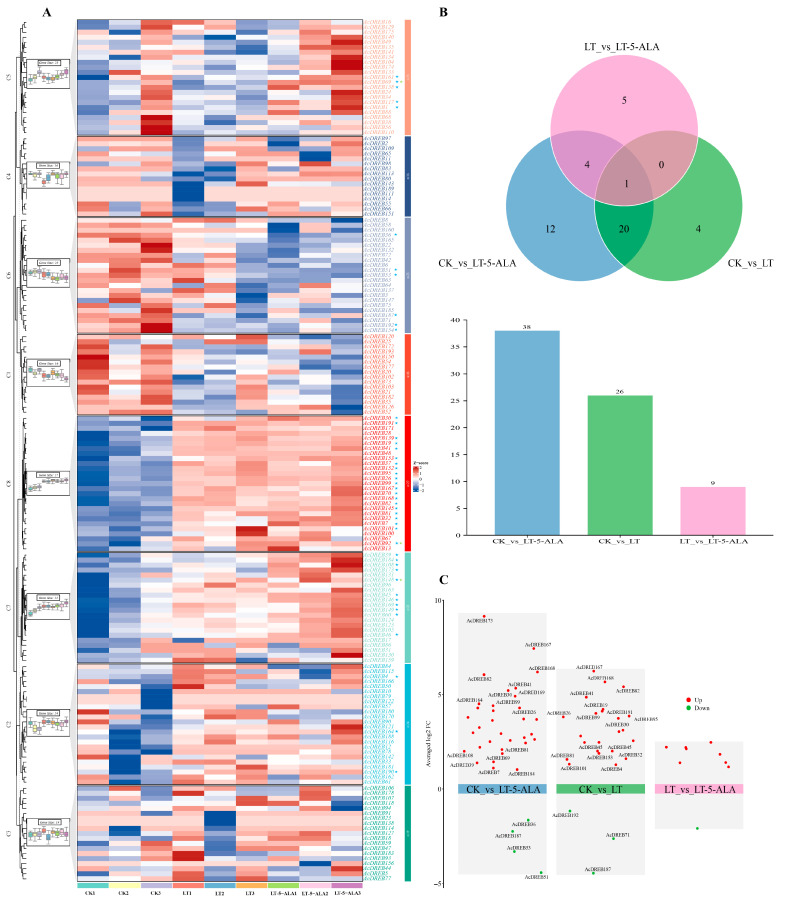
Heatmap and gene coexpression cluster analysis of kiwifruit DREB genes under the CK, LT, and LT-5-ALA treatments. (**A**) Cluster analysis. C1 to C8 in the diagram represent different clusters; the left side represents the gene expression patterns of different gene coexpression clusters; the middle side represents the heatmap; and the boxes in different colors represent different log2 (FPKM) values. The expression level gradually increased from blue to white and then to red. Genes and numbers are shown on the right, with blue pentagrams indicating differentially expressed genes in the AcDREB gene and green pentagrams indicating genes appearing in the KEGG pathway; n represents the number of genes in each cluster. (**B**) Shared and unique DEGs between different groups. (**C**) Scatter plot of differentially expressed genes, with the vertical axis of the graph showing the threshold of a 2-fold expression difference; red dots indicate significantly upregulated genes, and green dots indicate significantly downregulated genes.

**Figure 10 plants-14-02560-f010:**
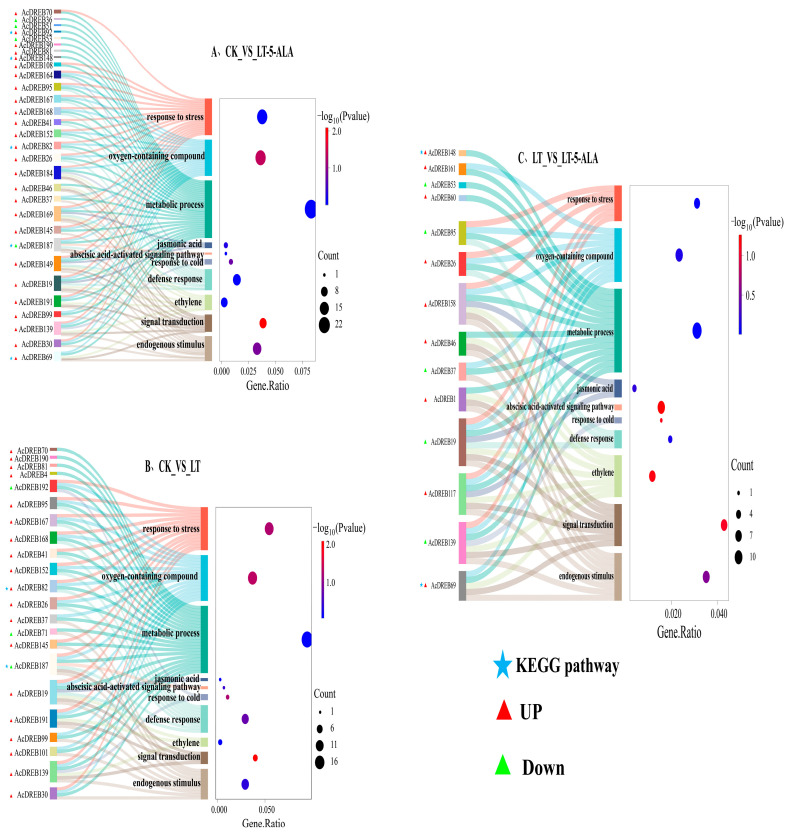
Functional enrichment Sankey plot of genes significantly differentially expressed in AcDREB that were screened and related to stress, including low-temperature stress, in kiwifruit seedlings regulated by exogenous 5-ALA. The genes are shown on the left vertical axis; the corresponding function is on the middle vertical axis. Data in black font correspond to the number of enriched genes; on the right is the bubble chart drawn using the GeneRatio and *p*-value; blue five-pointed stars denote genes in the KEGG pathway. The red triangles represent upregulated DEGs, and the green triangles represent downregulated DEGs. (**A**) CK_vs_LT-5-ALA group. (**B**) CK_vs_LT group. (**C**) LT vs. LT-5-ALA groups.

**Figure 11 plants-14-02560-f011:**
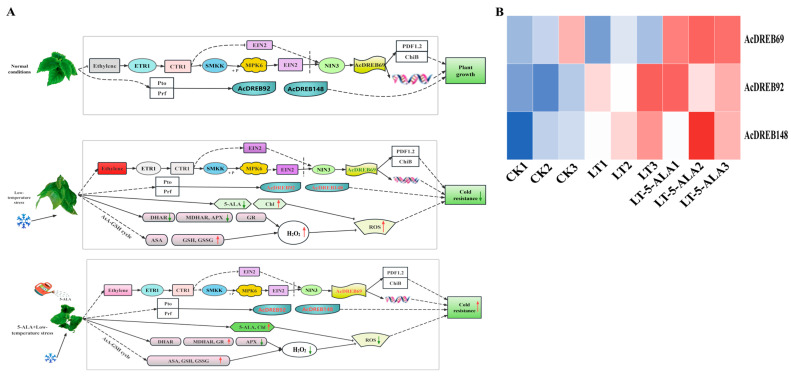
Model of the involvement of the three AcDREB genes in kiwifruit seedlings under low-temperature stress regulated by exogenous 5-ALA. (**A**) KEGG pathway diagram of plant hormone signaling pathways, MAPK signaling, and plant–pathogen interactions [45]. Concerning the ethylene signaling regulatory mechanism [46], the arrow in the figure indicates that the former item activates the latter item, and a non-arrowhead illustrates that the former item inhibits the latter item. The red arrow represents an increase, and the green arrow represents a decrease. The solid line represents the direct effect. There is no other process in the middle. However, the dotted line represents the indirect effect, with multiple action processes that have been omitted. The AcDREB gene was upregulated (red) and downregulated (green). (**B**) Expression patterns of the three DEGs involved in the KRGG pathway in the AcDREB gene.

**Figure 12 plants-14-02560-f012:**
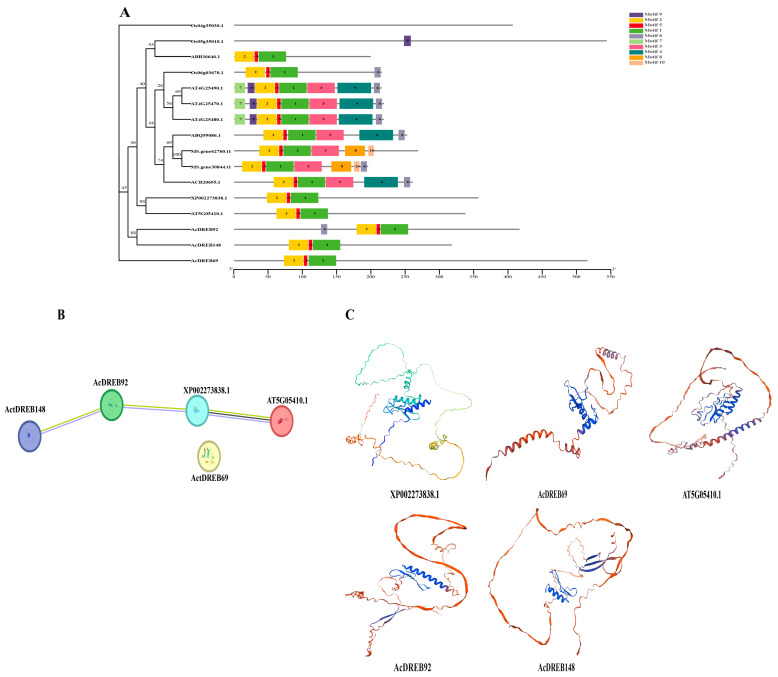
Phylogenetic relationships and conserved motif analysis of cold resistance genes and key DREB gene family genes. (**A**) The phylogenetic tree is on the left, and the distribution of conserved motifs is shown on the right. The motifs are represented by colored boxes, and the gray line represents the relative length of the proteins. (**B**) Protein–protein interaction network diagram of homologous genes, with each circle representing a protein. (**C**) Three-dimensional protein structures of homologous genes; the blue regions of the proteins in the figure are similar, and the analysis confidence is greater than 95%.

**Figure 13 plants-14-02560-f013:**
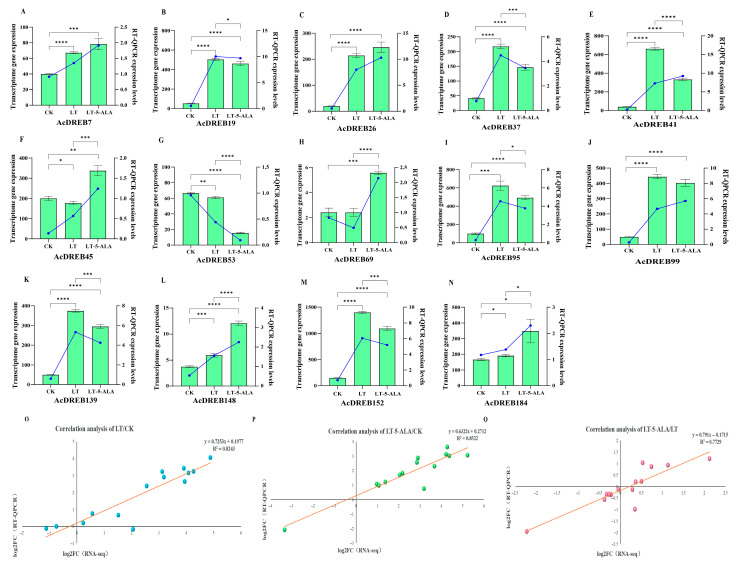
(**A**–**N**) show the transcriptome expression level and Rt-qPCR expression of the 14 AcDREB genes under CK, LT, and LT-5-ALA; (**O**–**Q**) represent the log2FC (RNA-seq) and log2FC (qRT-PCR) of the 14 AcDREB genes; and the relative expression level of AcActin (*Actinidia04464*) was used as an internal reference gene. All the data are the average of three biological replicates. The green bar graph illustrates the expression levels of the corresponding genes determined via transcriptome sequencing, and the blue line represents the relative expression detected via qRT-PCR. The vertical line on the histogram is the standard deviation line. * represents the *p*-value, and more * marks represent more significant differences.

## Data Availability

All the data generated or analyzed during this study are included in this published article. Kiwifruit genome annotation files are available at https://kiwifruitgenome.atcgn.com/#/home. The RNA-Seq data for exogenous 5-ALA mitigation under low-temperature stress can be found under registry no. PRJNA1208675. RNA-Seq data are publicly available at the National Center for Biotechnology Information. Additional data presented in this study are available in the Appendix A. All experimental studies and experimental materials involved in this study were performed in full compliance with relevant institutional, national, and international guidelines and regulations.

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
