# Peer review of "Genome-Wide Identification of DREB Gene Family in Kiwifruit and Functional Characterization of Exogenous 5-ALA-Mediated Cold Tolerance via ROS Scavenging and Hormonal Signaling"

_plants, 2025, doi:10.3390/plants14162560_

Round 1
Reviewer 1 Report
Comments and Suggestions for Authors
Dear Author
The manuscript by Tian et al identify the DREB Gene Family and explore the function of the DREB gene family in kiwifruit and the application of exogenous 5-ALA in alleviating low-temperature stress. The topic is of general interest; However, I have some major comments that are listed below
- Figure 2 the neighbor-joining (NJ) method with specific default settings was used. NJ is not a robust method for phylogenetic analysis in publication. The author should consider ML or more robust methods. An outside group should be considered for a stringent tree.
- Figure 8 The author declared that the most suitable concentration of exogenous elements was 30 mL/L 5-241 ALA. Why choose 30ml/L 5-MA? The author should justify concentration. For low temperature stress, people commonly use -4 degrees that may give an obvious phenotype. The author suggested that under low-temperature stress, exogenous 5-ALA can significantly relieve the degree of wilting and frostbite of kiwifruit seedlings while it is not obvious in Figure 8A. Does 30ml/L 5-MA cause damage to kiwi? The author should include the WT+5-MA group for control?
- In addition, under -2 degrees, the LT group has relatively lower 5-MA concentration compared to CK. However, when applying 30ml/l 5-MA to LT, the concentration increased slightly. Does that indicate the 5-MA treatment not working? Any idea of how 5-MA enter plants?
- The author measured the Enzyme (DHAR, MDHAR, and GR) activities and the contents of antioxidant substances (ASA, GSH, and GSSG). How are these enzymes connected to low tolerance?
- Line 686 the description is confusing. The transcriptome data of kiwifruit under control exogenous 5-ALA (LT5-ALA) treatments before low-temperature stress. is it better to have CK+5-ALA instead of LT5-ALA treatments before low-temperature stress.
- Figure 9A Heatmap indicated log2(FPKM) values. How does the data normalize? It looks like the three replications are not consistent. Whether the treatment or time of treatment is not working? The key take-away is whether these 193 AcDREB genes are up- or down- regulated in CK vs LT. Or whether these genes are induced by 5-ALA treatment? (CK VS CK+5-ALA). The author should consider reanalyzing the RNA seq data.
- Figure 11 the prediction model should go to discussion section.
- Figure 12 the conclusion cannot be drawn ‘ The results revealed that the phylogenetic clades of AcDREB69, ABB36646.1, and AT5G05410.1 were relatively close and contained the same conserved motifs, indicating they may have the same function (Figure 12A).’
- Figure 13 The author should explain the logic of these RT-PCR expressions analysis. How to explain the (ABCDEIJKLM) genes induced under LT, after applying 5-ALA, the genes expression reduced. FGHN genes expression remains same under LT, and induced by 5-ALA.
- Figure 13 The author should replace the Chinese character with English in the figure o p q.
- The author should consider rewriting the discussion. How are current data connect each other and contribute to the LT tolerance story.
Reviewer 2 Report
Comments and Suggestions for Authors
See attached file.

The paper need an extensive English editing.
Reviewer 3 Report
Comments and Suggestions for Authors
Dear Authors,
I have reviewed the manuscript. The manuscript focuses on the genome-wide identification of the DREB gene family in kiwifruit and the analysis of how exogenous 5-aminolevulinic acid (5-ALA) mitigates low-temperature stress. This is a highly relevant topic given the agricultural and economic importance of kiwifruit and the need to improve its resilience to abiotic stress, particularly under climate change conditions.
The novelty of the study lies in the integration of bioinformatics analyses with physiological experiments, providing the first functional insight into how DREB genes respond to exogenous 5-ALA under cold stress in kiwifruit. This offers valuable new knowledge that could guide breeding for cold-tolerant varieties.
Here’s a section-by-section summary with comments:
Introduction
The introduction gives solid background but is somewhat overloaded with general stress physiology details. I recommend tightening the focus on DREB genes in kiwifruit and stating the hypothesis more explicitly at the end.
Materials and Methods
The methods are comprehensive and detailed, but some parts (especially bioinformatics tools and parameters) could be shifted to supplementary material to improve readability. Adding clear justification for the choice of specific experimental conditions (e.g., 5-ALA concentration) would strengthen this section.
Results
The results are rich and cover many analyses, but they are sometimes overwhelming and repetitive, especially in describing gene family characteristics. I recommend condensing the presentation, focusing on the most biologically significant findings, and avoiding overloading with descriptive details.
Discussion
This section is underdeveloped. While the authors summarize the findings, they do not provide sufficient mechanistic interpretation or critical comparison with recent literature. I strongly recommend rewriting the discussion to deepen the analysis, integrate more recent studies, and clearly articulate how the findings advance the field.
Conclusions
The conclusions restate the main results but lack clear, practical take-home messages. I suggest reformulating this section to provide sharper recommendations for application or future research directions.
Reviewer 4 Report
Comments and Suggestions for Authors
The manuscript by Tian et. al. entitled “Genome-wide identification of the DREB gene family and model and functional analysis of kiwifruit seedlings under low-temperature stress relieved by exogenous 5-ALA” reports that ‘One hundred and ninety-three DREB gene families on 29 chromosomes were identified in kiwifruit’. The authors claim that ‘the use of exogenous 5-ALA could inhibit the accumulation of substances such as APX and DHAR and promote an increase in chlorophyll and the accumulation of 5-ALA, enzyme activities, and substances such as MDHAR, GR, ASA, GAH, and GSSH, further accelerating the removal of ROS and enhancing the cold tolerance of kiwifruit’. The authors conclude that the study of AcDREB genes under low-temperature stress regulation by exogenous 5-ALA provides a basis for further studies on the kiwifruit breeding. The work is technically sound piece of research, and within the scope of Plants, but it requires minor revision before its acceptance for publication.
- Provide more references that related to respond of abiotic stresses such as cold, drought, heat, salinity, and waterlogging on kiwifruit plant
- Rewrite Title to reflect the contents, rewrite Abstract to emphasize main points
- Simplify the text contents (reviewer think that 30% more can be removed) to make clear clues to attract the readers interesting.
need to be simplified
Round 2
Reviewer 1 Report
Comments and Suggestions for Authors
The author has addressed all my previous comments
Author Response
Comments: [The author has addressed all my previous comments]
Response: Thank you very much for your previous comments on my paper and for agreeing to publish it.
Reviewer 2 Report
Comments and Suggestions for Authors
The manuscript still needs a lot of improvement.
- Extensive English editing by a native speaker or via MDPI.
- A functional study is needed such as yeast two hybrid for a potential DREBs Transcription factor.
- Many errors types mainly between word and references and also in the texte of manuscript (at references section)
- I insiste that the result section must be presented directly after the introduction section.
- Please refer to the journal instruction for the presentation of yours references.
Comments on the Quality of English Language
Extensive English editing is needed.
Author Response
请参阅附件

Reviewer 3 Report
Comments and Suggestions for Authors
Thank you, I recommand it for publication.
Author Response
Comments: [Thank you, I recommand it for publication. ]
Response: Thank you very much for agreeing to the publication of my thesis.
Round 3
Reviewer 2 Report
Comments and Suggestions for Authors
The manuscript still need improvement.
- Many errors types still exist mainly between word and references. This point need to be considered.
- English editing language is necessary.
- Please respect the instruction of the journal in the order of sections. This point is very important (Abstract; Introduction; Results; Discussion; Materials and Methods; Conclusions; References). Please take care for the order of references after changing the order of these sections.
- I'm not convinced with the response of the authors in relation with the functional studies of DREB genes. I consider that it is very important for the validation of your bioinformatic results. I suggest at least two hybrid experiments.
The quality of English is poor and I suggest an Enlish editing by MDPI service.
Round 4
Reviewer 2 Report
Comments and Suggestions for Authors
Authors responded to all required points.